# A Surrogate Perspective on Convergence of Fixed-Target DQN

## Abstract

Deep Q-Networks (DQNs) approximate value iteration by freezing a target network, forming Bellman-optimality targets, and running an inner regression loop that minimizes a smooth *surrogate loss* (e.g., MSE/Huber loss) on data from a replay buffer. Yet, control is governed by the Bellman optimality operator's $\gamma$-contraction in $\ell_\infty$, whereas common objectives (MSE/Huber loss) measure progress in an averaged $\ell_2/\ell_1$ geometry; thus, improving the surrogate need not reduce the worst-case Bellman error that drives performance. To bridge this gap we connect the progress in the inner loop for each surrogate to the Bellman residual under the $\ell_\infty$ norm. Our analysis provides explicit thresholds denoted by $\alpha$ under which sufficient progress guarantees a $\ell_\infty$ contraction up to a standard approximation floor. This yields new convergence guarantees for DQN with fixed targets trained using MSE or Huber-type losses, and motivates a novel soft-$\ell_\infty$ surrogate that smoothly approximates the sup-norm and better matches the $\ell_\infty$ geometry resulting in the least stringent inner loop accuracy requirements. Across Atari and MuJoCo benchmarks, soft-$\ell_\infty$ consistently reduces the worst-case Bellman residual and matches or improves returns relative to standard objectives, providing a practical, geometry-aligned recipe for stable DQN training.

## 1. Introduction

Deep Q-Networks (DQNs) (Mnih et al., 2015) achieve strong empirical performance across a wide range of high-dimensional control tasks, yet *when and in what sense DQN-style Q-learning converges with function approximation* remains only partially understood. A central challenge is that practical stability relies on a collection of interacting de-

sign choices, most notably experience replay (Lin, 1992) and a target network that is held fixed for many inner-loop updates (Mnih et al., 2015); these mechanisms are widely viewed as essential for mitigating the instabilities that arise from combining function approximation, bootstrapping, and off-policy learning (the "deadly triad") (Sutton & Barto, 1998; Van Hasselt et al., 2018). In this work, we study DQN through the lens of *fixed-target fitted Q-iteration*: an outer loop that applies an (approximate) Bellman optimality update using a frozen target, and an inner loop that approximately solves a supervised regression subproblem on replay data (Ernst et al., 2005a;b; Fan et al., 2020).

Bootstrapping is notoriously delicate under function approximation: classical TD and Q-learning updates are not true gradient descent on a global objective, and can diverge even in linear settings (Baird et al., 1995; Van Hasselt et al., 2018). In the *linear policy-evaluation* setting, least-squares TD methods (e.g., LSTD/LSPE) compute an approximate value function by solving a *projected* Bellman fixed-point equation in a feature subspace (Bradtke & Barto, 1996; Bertsekas & Ioffe, 1996; Bertsekas, 2009). These projected-equation viewpoints can be formalized as *variational inequalities*, enabling stability analyses via monotonicity in weighted Euclidean geometries for policy evaluation (Bertsekas, 2009).

However, *control* is governed by a different geometry: the Bellman optimality operator is a $\gamma$-contraction in the sup norm (Szepesvári, 2022; Bertsekas & Tsitsiklis, 1996), and sup-norm approximation error directly controls worst-case performance guarantees for greedy policies (Bertsekas & Tsitsiklis, 1996). In contrast, the inner regression problems optimized in DQN (e.g., MSE or Huber-type objectives on replay data) measure progress in distribution-weighted $\ell_2/\ell_1$ geometries, so descending the regression loss need not certify a decrease in the *worst-case* Bellman residual relevant to control (Munos, 2007). We address this gap by developing a *geometry-aware surrogate methodology* for fixed-target Q-learning. Our approach uses the $\alpha$-*descent* condition from D'Orazio et al. (2025), and extends it to the sup-norm geometry as well as explicitly handling the geometry mismatch between the surrogate loss and the sup-norm. By explicitly quantifying the mismatch between a training loss's geometry and sup-norm, we derive surrogate-specific thresholds $\alpha$ under which sufficient inner-loop improvement guarantees an $\ell_\infty$ contraction of the Bellman error (up to

[1]Anonymous Institution, Anonymous City, Anonymous Region, Anonymous Country. Correspondence to: Anonymous Author <anon.email@domain.com>.

Preliminary work. Under review by the International Conference on Machine Learning (ICML). Do not distribute.

a standard approximation floor under coverage and realizability assumptions). This yields convergence guarantees for fixed-target DQN trained with MSE and Huber-type losses and further clarifies why freezing the target network can be sufficient to stabilize DQN. The same analysis also motivates a temperature-controlled soft-$\ell_\infty$ surrogate that interpolates between quadratic and sup-norm regimes while imposing the least stringent inner-loop accuracy requirement among the surrogates we consider. Empirically, on Atari and MuJoCo benchmarks, this soft-$\ell_\infty$ objective consistently reduces the worst-case Bellman residual and matches or improves return relative to standard surrogate losses (i.e., MSE/Huber loss), illustrating how our methodology can guide principled choices in DQN training.

## 2. Background

### 2.1. Notations and Definitions

For a finite set $\mathcal{S}$, let $\Delta_\mathcal{S} := \{p \in \mathbb{R}^{|\mathcal{S}|} : p \geq 0, \sum_{s \in \mathcal{S}} p(s) = 1\}$. For $x \in \mathbb{R}^n$, $\|x\|_\infty := \max_i |x_i|$. Given a norm $\|\cdot\|$, its dual is $\|y\|_* := \sup_{\|x\| \leq 1} \langle y, x \rangle$. We denote the gradient of a differentiable function $f : \mathbb{R}^n \to \mathbb{R}$ as $\nabla f(x) := (\partial_1 f(x), \ldots, \partial_n f(x))^\top$, where $\partial_i f(x) := \frac{\partial f(x)}{\partial x_i}$ is the $i$-th partial derivative.

**Definition 2.1** (Lipschitzness, Smoothness)**.** We say that a (differentiable and convex) function $f : \mathcal{X} \to \mathbb{R}$ is respectively $L$-Lipschitz, $L$-smooth if for any $x, x' \in \mathcal{X}$ it satisfies,

$$|f(x) - f(x')| \leq L \|x - x'\|, \quad (1)$$

$$\|\nabla f(x) - \nabla f(x')\|_* \leq L \|x - x'\|, \quad (2)$$

**Definition 2.2** (Polyak–Łojasiewicz (PL) condition (Karimi et al., 2016))**.** Let $f : \mathcal{X} \to \mathbb{R}$ be differentiable on $\mathcal{X}$, and write $f_\star := \inf_{x \in \mathcal{X}} f(x)$. We say that $f$ satisfies the PL condition with parameter $\mu > 0$ if, for all $x \in \mathcal{X}$,

$$\frac{1}{2} \|\nabla f(x)\|_*^2 \geq \mu(f(x) - f_\star).$$

### 2.2. Reinforcement Learning

We consider an infinite-horizon discounted Markov decision process (MDP) $(\mathcal{S}, \mathcal{A}, P, r, \gamma)$, with state space $\mathcal{S}$, finite action set $\mathcal{A}$, transition kernel $P(\cdot \mid s, a)$, bounded rewards $|r(s, a)| \leq R_{\max}$, and discount $\gamma \in (0, 1)$. A policy $\pi(\cdot \mid s) \in \Delta_\mathcal{A}$ induces the state-value function $V^\pi(s) = \mathbb{E}_\pi\left[\sum_{t \geq 0} \gamma^t r(s_t, a_t) \mid s_0 = s\right]$ and action-value function $Q^\pi(s, a) = \mathbb{E}_\pi\left[\sum_{t \geq 0} \gamma^t r(s_t, a_t) \mid s_0 = s, a_0 = a\right]$.

Dynamic programming characterizes $Q^\pi$, and the optimal action-value function $Q^\star(s, a) = \max_\pi Q^\pi(s, a)$, as unique fixed points of their respective Bellman operators. For any policy $\pi$, the Bellman evaluation operator is denoted by $T^\pi$ and the Bellman optimality operator $T$ is

$$(TQ)(s, a) = r(s, a) + \gamma \mathbb{E}_{s' \sim P(\cdot \mid s, a)}\left[\max_{a' \in \mathcal{A}} Q(s', a')\right].$$

Both $T^\pi$ and $T$ are $\gamma$-contractions in the sup norm $\|\cdot\|_\infty$; hence according to Banach's fixed-point theorem each admits a unique fixed point (respectively $Q^\pi$ and $Q^\star$), and the value iteration algorithm which implements updates over $k$ steps as $Q_{k+1} = TQ_k$ converges geometrically in $\|\cdot\|_\infty$ to $Q^\star$ (Bellman, 1957; Bertsekas, 1987). This contraction property motivates value based methods that implement Bellman updates via fitted regression to Bellman targets, including Neural Fitted Q-Iteration (Riedmiller, 2005) and Deep Q-Networks (DQN) (Mnih et al., 2013).

Deep Q-learning approximates value iteration via fitted regression to Bellman targets. In DQN (Mnih et al., 2013), at outer iteration $t$ a target network $Q_{\hat{\theta}_t}$ is frozen and kept fixed for $K_{\text{target}}$ updates of the online network. Given a transition $(s, a, r, s')$ sampled from the replay buffer, the (one-step) Bellman target is $Y_t(s, a, s') = r + \gamma \max_{a' \in \mathcal{A}} Q_{\hat{\theta}_t}(s', a')$, and the inner-loop objective is

$$\ell_t(\theta) = \mathbb{E}_{(s,a,r,s') \sim \mathcal{D}}[\mathcal{L}(Q_\theta(s, a) - Y_t(s, a, s'))], \quad (3)$$

where $\mathcal{L}$ is an $L$-smooth surrogate loss (e.g., MSE, Huber). Despite the $\ell_\infty$-*contraction* property of $T$ that governs control, minimizing $\ell_t$ is typically carried out in an averaged geometry induced by the surrogate, which does not automatically imply $\ell_\infty$-contraction. To bridge this gap, we formalize a sufficient condition on the surrogate decrease under which the outer iteration contracts in $\ell_\infty$. The next section introduces the surrogate loss optimization framework and the $\alpha$-conditions that can help us translate per-step surrogate improvement into an explicit $\ell_\infty$-contraction guarantee.

### 2.3. Surrogate methods

The iterative construction and minimization of tractable surrogate objectives traces back to the majorization–minimization (MM) paradigm (Mairal, 2015), with the Expectation–Maximization (EM) algorithm (Dempster et al., 1977) as a canonical example. In reinforcement learning, the same principle underlies modern policy optimization: methods such as TRPO (Schulman et al., 2015) and PPO (Schulman et al., 2017) optimize surrogate losses that approximate return improvement while enforcing trust-region style stability. Recent work unifies these ideas through a functional-mirror-ascent perspective, clarifying when such surrogates guarantee monotone improvement independent of parameterization (Vaswani et al., 2021)

Surrogate methods replace direct optimization in parameter space $\Theta$ with a sequence of auxiliary subproblems that drives the predictions toward a target update in prediction space $\mathcal{Z}$, enabling convergence to be stated at the level of

predictions rather than in the (potentially) nonconvex parameter landscape. We consider the composition of continuous objectives $\min_{\theta \in \Theta} \ell(g(\theta))$, with realizable set $\mathcal{Z} := \mathrm{cl}\{g(\theta) : \theta \in \Theta\}$, so that $\inf_{\theta \in \Theta} \ell(g(\theta)) \equiv \inf_{z \in \mathcal{Z}} \ell(z)$. At outer step $t$, starting from $z_t = g(\theta_t)$, we form a *target* $y_t \in \mathcal{Z}$ that approximates a prediction-space step, and then update parameters by minimizing the *surrogate loss*

$$\ell_t(\theta) = \mathcal{L}(g(\theta) - y_t),$$

where $\mathcal{L}$ is a smooth convex discrepancy (e.g., mean squared error, KL divergence) chosen to match the task geometry. This makes parameter space updates follow the well-conditioned descent direction in the prediction space, where convergence guarantees are available. Prediction space often exhibits simpler geometric structure (e.g., convexity/monotonicity) whereas expressive nonlinear parameterizations distort this geometry and render the objective nonconvex. Surrogates bridge this gap by specifying a well-behaved target in prediction space and pulling the model toward it through these subproblems.

D'Orazio et al. (2025) extend the surrogate perspective to general variational inequalities (VIs) and obtain *global* prediction-space convergence even when the inner loop solves each surrogate only *approximately*. Their analysis assumes that the (hidden) VI operator is $L$-smooth and $\mu$-strongly monotone in an $\ell_2$ geometry, and constructs a one-step prediction-space target corresponding to a (projected) gradient step; the inner loop then fits model predictions to this target by minimizing a squared-error surrogate. Progress is certified via an $\alpha$-descent condition on the surrogate subproblem (our Eq. 8); under standard step-size choices and the smoothness/strong-monotonicity assumptions, this condition implies a *linear* contraction of the prediction iterates $z_t = g(\theta_t)$ toward the VI solution. DQN fits the same template: it freezes a target network to define the bootstrapped TD targets and then performs fitted regression to Bellman targets for a fixed number of updates by minimizing an $\ell_1/\ell_2$-type surrogate loss (e.g., squared loss or Huber) on replay samples. However, existing VI-style surrogate guarantees do not directly transfer to control.

Unlike the case in D'Orazio et al. (2025), the Bellman optimality operator is not a contraction in a weighted $\ell_2$ norm or an inner-product norm but in the $\ell_\infty$ norm. Therefore the averaged $\ell_1/\ell_2$-type surrogates do not naturally fit the geometry, and more importantly, existing analyses using strong monotonicity arguments do not apply. Alternatively, a natural loss that directly aligns with control is the $\ell_\infty$ Bellman residual (Williams & Baird, 1993) but is impractical to minimize in practice via a surrogate approach. Consequently, in the DQN case with $\ell_1/\ell_2$-type surrogates a separate geometry-alignment argument is required: reducing an averaged regression surrogate does not, by itself, ensure a decrease of the $\ell_\infty$ Bellman residual that governs con-

traction and greedy-policy improvement. More importantly, if the $\ell_\infty$ Bellman residual is essential for guaranteeing greedy-policy improvement but is not aligned with standard surrogates then what losses are best suited for this task?

## 2.4. Surrogate Losses for Q learning

Motivated by the mismatch between the regression geometry and the control-relevant $\ell_\infty$ metric, we instantiate a family of surrogate losses on the Bellman residual. The empirical objective is naturally written in terms of sampled TD error $Q_\theta(s, a) - Y_t(s, a, s')$. To avoid complications that naturally arise from studying this sample-based objective, we replace $Y_t$ by its conditional expectation and work with the corresponding population residual $\varepsilon_{\theta,t} := Q_\theta - TQ_{\hat\theta_t}$. We show in Section 4 that our performance claims are validated by experiments with the empirical error. Let $w$ denote the replay-induced distribution over $\mathcal{S} \times \mathcal{A}$. We study three smooth surrogates with increasing emphasis on worst-case discrepancies: MSE, a soft-Huber proxy (log-cosh), and a soft-$\ell_\infty$ proxy (log-sum-cosh).

**(a) MSE (Mean Squared Error).** A common choice of surrogate loss in DQN and its variants (Mnih et al., 2015; Sutton & Barto, 1998) is the squared TD error:

$$\mathcal{L}_{\mathrm{MSE}}(\varepsilon_{\theta,t}) = \tfrac{1}{2}\mathbb{E}_{(s,a)\sim w}\big[\varepsilon_{\theta,t}(s,a)^2\big]. \quad (4)$$

MSE measures Bellman residual in an $\ell_2$ geometry and prioritizes average error reduction. While analytically convenient under standard light-tailed noise assumptions (Huber, 2011), its quadratic penalty disproportionately up-weights rare large TD errors, so a small number of outliers can dominate the updates. Practical implementations commonly temper squared-error updates by using a clipped TD-error objective (Mnih et al., 2015). This practice naturally motivates robust surrogates such as the Huber loss.

**(b) Huber and Soft Huber (Log Cosh).** A standard robust alternative to MSE is Huber loss with threshold $\delta > 0$:

$$\mathcal{L}_\delta(u) = \begin{cases} \tfrac{1}{2}u^2, & |u| \le \delta, \\ \delta\big(|u| - \tfrac{1}{2}\delta\big), & |u| > \delta, \end{cases}$$

which is quadratic near 0 and linear in the tails, thereby bounding the per-sample influence (Huber, 1964). For theoretical clarity and smooth optimization, we use a log-cosh proxy (soft Huber) (Saleh & Saleh, 2022):

$$\mathcal{L}_{\mathrm{LC}}(\varepsilon_{\theta,t}, \beta) = \tfrac{1}{\beta^2}\mathbb{E}_{(s,a)\sim w}\big[\mathrm{logcosh}\big(\beta\,\varepsilon_{\theta,t}(s,a)\big)\big], \quad (5)$$

where $\beta > 0$ is a temperature parameter, with an effective Huber threshold $\delta \approx 1/\beta$. This surrogate is convex and infinitely differentiable, its per-sample derivative with respect to the TD error is $\tanh(\beta(\varepsilon_{\theta,t}))/\beta$, which is bounded between $[-\tfrac{1}{\beta}, \tfrac{1}{\beta}]$; hence, large TD errors are naturally down-weighted. The temperature $\beta$ controls the transition scale:

smaller $\beta$ yields a wider MSE-like (quadratic) regime, while larger $\beta$ induces earlier saturation and more $\ell_1$-like tail behavior, improving robustness to heavy-tailed targets. Robust Bellman-error objectives have been proposed as principled replacements for MSE and analyzed as more stable alternatives under function approximation (Patterson et al., 2022). Related empirical work has also incorporated Huber-style losses for robustness in offline RL with data corruption, and some applied DRL systems to tune the Huber threshold during training using heuristic schedules (Yang et al., 2023; Xu et al., 2025). In contrast, our soft-Huber temperature $\beta$ provides a differentiable control knob that can be adapted within a unified surrogate framework, enabling convergence guarantees that explicitly connect robust regression behavior to our control-relevant analysis.

**(c) Soft $\ell_\infty$ (Log Sum Cosh).** The Bellman optimality operator contracts in the non-smooth $\ell_\infty$ geometry, therefore directly optimizing an $\ell_\infty$ residual leads to a non-smooth objective and can be challenging for standard first-order methods. Following classical max-structure smoothing (Nesterov, 2005), we replace $\|\cdot\|_\infty$ with the smooth proxy:

$$\mathcal{L}_{\text{LSC}}(\varepsilon_{\theta,t}, \beta) = \tfrac{1}{\beta} \log \mathbb{E}_{(s,a)\sim w}\big[\cosh\big(\beta\varepsilon_{\theta,t}(s,a)\big)\big]. \quad (6)$$

As $\beta \to \infty$, $\mathcal{L}_{\text{LSC}}$ concentrates on the largest coordinates and approximates $\|\varepsilon_{\theta,t}\|_\infty$ up to an $O(1/\beta)$ additive term (See Lemma B.3). For $|\beta\varepsilon_{\theta,t}| \ll 1$, it is locally quadratic (MSE-like up to a $\beta$-dependent scaling), yielding a more uniform aggregation across coordinates. Thus, increasing $\beta$ provides a smooth transition from a well-conditioned, uniformly-weighted regime to a worst-case focused, $\ell_\infty$-aligned regime that is directly tied to Bellman contraction.

Moreover, $\mathcal{L}_{\text{LSC}}$ adaptively emphasizes larger discrepancies. Let $A(\varepsilon_{\theta,t}) := \sum_j w_j \cosh\big(\beta(\varepsilon_{\theta,t})_j\big)$ and $p_i := \frac{w_i \cosh\big(\beta(\varepsilon_{\theta,t})_i\big)}{A(\varepsilon_{\theta,t})}$ (a softmax-like weight over coordinates). Then $\nabla\mathcal{L}_{\text{LSC}}(\varepsilon_{\theta,t})_i = p_i \tanh\big(\beta(\varepsilon_{\theta,t})_i\big)$, indices with larger $|(\varepsilon_{\theta,t})_i|$ receive greater weight $p_i$. This is conceptually reminiscent of error-based prioritization (Schaul et al., 2015), but implemented as a differentiable reweighting within the loss rather than a sampling heuristic.

## 3. Main Contribution

In this section, we analyze the standard DQN two-loop procedure from a fitted Q-iteration viewpoint. At each outer iteration $t$, we freeze a target network, build Bellman optimality backups, and run $K$ inner updates that regress to the Bellman targets by minimizing a smooth surrogate residual. Our goal is to identify when surrogate descent in the inner loop yields monotone progress in the control relevant $\ell_\infty$ norm towards $Q^\star$, despite the mismatch between the $\ell_\infty$ geometry underlying Bellman optimality contraction and the averaged geometry of common surrogates. We formalize

inner loop progress via an $\alpha$-condition, requiring a multiplicative decrease of the surrogate residual by $\alpha \in [0,1)$. Under standard (approximate) Bellman completeness and coverage assumptions, we derive surrogate-specific admissible $\alpha$ ranges that imply outer loop $\ell_\infty$ contraction up to an approximation floor controlled by inherent Bellman error (see Definition 3.1). Section 3.1 presents the main theorem linking inner loop optimization progress to outer loop control progress. For each surrogate loss, we provide an explicit $\alpha$-condition and the corresponding asymptotic error floor. Section 3.2 provides step complexity guarantees by translating the required $\alpha$-condition into a sufficient inner-loop step budget $K$. Section 3.3 discusses the resulting algorithm.

### 3.1. $\alpha$-conditions for Surrogate Losses

We now establish a sufficient condition under which progress of the inner regression step implies uniform improvement towards $Q^\star$ in the control relevant $\ell_\infty$ norm. We interpret the outer loop as an inexact fitted Q-iteration step and quantify the impact of (i) inner-loop optimization error and (ii) function-class approximation error. To translate regression progress into control progress, we impose two standard conditions.

**Definition 3.1** (Inherent Bellman Error (Munos & Szepesvári, 2008)). Let $\mathcal{Z} = \{Q_\theta : \theta \in \Theta\}$ denote the action-value function class and $T$ be the Bellman optimality operator. The inherent Bellman error is:

$$\varepsilon_\infty := \sup_{Q\in\mathcal{Z}} \inf_{Q'\in\mathcal{Z}} \|Q' - TQ\|_\infty. \quad (7)$$

The special case $\varepsilon_\infty = 0$ corresponds to *Bellman completeness*, i.e., $TQ \in \mathcal{Z}$ for all $Q \in \mathcal{Z}$. In general, $\varepsilon_\infty$ measures the worst-case $\ell_\infty$-approximation error incurred when representing a target $TQ$ within $\mathcal{Z}$.

This assumption is standard in analyses of fitted Q iteration (Munos & Szepesvári, 2008; Neu & Okolo, 2023). To control how regression error measured under the sampling distribution $w$ transfers to uniform $\ell_\infty$ error, we additionally require a standard data-coverage condition on $w$.

**Assumption 3.2** (Sufficient coverage). Let $w$ be the sampling distribution over $\mathcal{S} \times \mathcal{A}$ used to define the inner-loop regression objective (i.e., expectations in the surrogate loss). Define the minimum sampling mass $\underline{w} := \inf_{(s,a)\in\mathcal{S}\times\mathcal{A}} w(s,a)$. We assume $\underline{w} > 0$ (i.e., $w$ has full support and is bounded away from zero).

With misspecification quantified by $\varepsilon_\infty$ and distribution mismatch controlled by Assumption 3.2, we can now state an $\alpha$-condition that specifies how much inner-loop surrogate decrease is sufficient to guarantee $\ell_\infty$ improvement in the outer iteration. Fix a surrogate $\text{sur} \in \{\text{MSE}, \text{LC}, \text{LSC}\}$ at step $t$, denote the minimum surrogate loss attainable by

our function approximator as $\mathcal{L}^{\star}_{\mathrm{sur},t}$ we say the inner loop achieves reduction factor $\alpha_{\mathrm{sur}} \in [0,1)$ if it returns $\theta_{t+1}$ satisfying,

$$\mathcal{L}_{\mathrm{sur}}\big(\varepsilon_{\theta_{t+1},t},\beta\big) - \mathcal{L}^{\star}_{\mathrm{sur},t}$$
$$< \alpha_{\mathrm{sur}}(\mathcal{L}_{\mathrm{sur}}\big(\varepsilon_{\theta_t,t},\beta\big) - \mathcal{L}^{\star}_{\mathrm{sur},t}). \quad (8)$$

We refer to Equation 8 as the $\alpha$-*condition* for sur at outer step $t$. The following theorem gives surrogate-specific admissible values of $\alpha$ under which the outer iterates contract linearly up to a surrogate-dependent error floor.

**Theorem 3.3** (Surrogate $\alpha$-conditions and $\ell_{\infty}$ contraction to a radius)**.** *Assume Definition 3.1 holds with small inherent Bellman error $\varepsilon_{\infty}$ and Assumption 3.2 holds. Fix a surrogate* sur $\in \{\mathrm{MSE}, \mathrm{LC}, \mathrm{LSC}\}$, *and let $(Q_t)_{t\geq 0} \subset \mathcal{Z}$ be the iterates produced by the outer loop. Suppose the inner loop at outer step $t$ satisfies the $\alpha$-condition Equation 8 for some $\alpha \in [0,1)$. Define $E_t := \|Q_t - Q^{\star}\|_{\infty}$. A sufficient condition for the outer update to be contractive in $\ell_{\infty}$ is that $\alpha$ lies in the surrogate-dependent range*

$$\alpha_{\mathrm{MSE}} < \underline{w}\left(\frac{1-\gamma}{1+\gamma}\right)^2, \; \alpha_{\mathrm{LC}} < \underline{w}\cdot\frac{1-\gamma}{1+\gamma}, \; \alpha_{\mathrm{LSC}} < \frac{1-\gamma}{1+\gamma}.$$

*In particular, choosing $\alpha_{\mathrm{MSE}} = \frac{\underline{w}}{4}\left(\frac{1-\gamma}{1+\gamma}\right)^2$, $\alpha_{\mathrm{LC}} = \frac{\underline{w}}{2} \cdot \frac{1-\gamma}{1+\gamma}$, and $\alpha_{\mathrm{LSC}} = \frac{1-\gamma}{2+2\gamma}$ yields the explicit bound*

$$E_t \leq \left(\frac{1+\gamma}{2}\right)^t E_0 + R_{sur},$$

*where the surrogate-dependent radii are*

$$R_{\mathrm{MSE}} = \frac{2}{1-\gamma}\sqrt{\frac{1}{\underline{w}} - \frac{(1-\gamma)^2}{4(1+\gamma)^2}}\,\varepsilon_{\infty},$$

$$R_{\mathrm{LC}} = \frac{2}{1-\gamma}\left(\left(\frac{1}{\underline{w}} - \frac{1-\gamma}{2(1+\gamma)}\right)\varepsilon_{\infty} + \frac{1}{\beta}\log 2\right),$$

$$R_{\mathrm{LSC}} = \frac{1+3\gamma}{1-\gamma^2}\varepsilon_{\infty} + \frac{2}{\beta(1-\gamma)}\log\frac{2}{\underline{w}}.$$

When $\varepsilon_{\infty} = 0$ (i.e., $\mathcal{Z}$ is *Bellman complete*), the approximation component of the radius disappears: $R_{\mathrm{MSE}} = 0$, and the remaining radii $R_{\mathrm{LC}}$ and $R_{\mathrm{LSC}}$ are purely due to surrogate smoothing, scaling as $O(1/\beta)$.

*Remark* 3.4 (Uniform, bias-free $\alpha$-condition)*.* In addition to the surrogate-specific admissible ranges in Theorem 3.3, one may enforce the more stringent MSE-style requirement

$$\alpha < w_{\min}\left(\frac{1-\gamma}{1+\gamma}\right)^2.$$

This yields the same $\ell_{\infty}$ contraction rate as the MSE analysis and removes the surrogate-smoothing bias for LC/LSC (no additive $O(1/\beta)$ term in the radius). Consequently the steady-state radius is purely approximation-driven and satisfies $R_{\mathrm{sur}}(\varepsilon_{\infty}) \to 0$ as $\varepsilon_{\infty} \to 0$ (in particular $R_{\mathrm{sur}} = 0$ under Bellman completeness).

Theorem 3.3 links *optimization progress* in the fitted regression step to *control-relevant* $\ell_{\infty}$ progress in the outer iteration. Two structural quantities govern this translation. First, the inherent Bellman error $\varepsilon_{\infty}$ measures the worst-case $\ell_{\infty}$ approximation incurred when representing Bellman backups $TQ$ within $\mathcal{Z}$; when $\varepsilon_{\infty} = 0$ the class is Bellman complete (closed under $T$) and the approximation-driven component of the radius vanishes. Second, the coverage parameter $\underline{w}$ controls how surrogate error measured *on average under* the replay distribution transfers to a *uniform* $\ell_{\infty}$ bound; smaller $\underline{w}$ corresponds to weaker coverage and amplifies the constants in the $\ell_{\infty}$ guarantee.

The admissible $\alpha$ thresholds reflect the mismatch between the surrogate geometry and the $\ell_{\infty}$ control geometry. For MSE and LC (soft-Huber), converting an averaged regression guarantee into a uniform bound introduces a dependence on $\underline{w}$, hence the stricter conditions $\alpha_{\mathrm{MSE}} = O\big(\underline{w}((1-\gamma)/(1+\gamma))^2\big)$ and $\alpha_{\mathrm{LC}} = O\big(\underline{w}(1-\gamma)/(1+\gamma)\big)$. In contrast, the LSC (soft-$\ell_{\infty}$) surrogate is a smooth proxy for a max-residual objective and therefore aligns more directly with $\ell_{\infty}$ control: its admissible range does not depend on $\underline{w}$. Finally, LC and LSC incur an additive $O(1/\beta)$ bias because a smooth max only approximates the nonsmooth $\ell_{\infty}$ residual; this term can be removed by imposing the stronger uniform $\alpha$-condition (Remark 3.4), at the cost of a sharper objective that can require more inner-loop work which will be quantified in the next section.

Within the admissible ranges, the radius $R_{\mathrm{sur}}$ increases monotonically with $\alpha$, tighter inner-loop optimization yields a smaller steady-state $\ell_{\infty}$ error ball, while looser inner loops inflate the radius and eventually violate contractivity once $\alpha$ exceeds the surrogate-specific threshold.

### 3.2. Inner Loop Analysis

Having established $\alpha$-conditions that guarantee an outer-loop $\ell_{\infty}$ contraction under sufficient surrogate decrease, we next quantify the number of gradient steps needed to achieve the $\alpha$-condition for each surrogate, with $R_t := \max_{0\leq k\leq K} \|Q_{\theta_{t,k}} - TQ_{\hat{\theta}_t}\|_{\infty}$ denoting the largest frozen-target Bellman residual encountered in the inner loop at outer step $t$[1]. This follows from the curvature each surrogate induces, which controls the inner-loop optimization complexity. Throughout this subsection we study the complexity in the *tabular* setting, i.e., we analyze updates directly on $Q$-values to obtain a clean illustrative proxy for step complexity. In contrast, with parameterized models the nonlinear map couples the optimization geometry with the parametrization, making comparable analyses substantially more delicate. We analyze the *normalized sign* update, the steepest-descent step in $\|\cdot\|_{\infty}$. For an $L_{\infty}$-smooth objective

---

[1]If $|r| \leq R_{\max}$, then both $\|Q\|_{\infty}$ and $\|TQ\|_{\infty}$ are bounded by the same constant $R_{\max}/(1-\gamma)$, so $R_t \leq 2R_{\max}/(1-\gamma)$.

$f$, the update is

$$\theta_{k+1} = \theta_k - \frac{\|\nabla f(\theta_k)\|_1}{L_\infty} \operatorname{sign}\big(\nabla f(\theta_k)\big). \qquad (9)$$

This update has been considered as a faithful proxy for the standard optimizers like AdamW (Balles et al., 2020; Orvieto & Gower, 2025). The step complexity is bounded by combining $\ell_\infty$-smoothness with an $\ell_1$-PL inequality that holds on an invariant sublevel set (see Lemma C.6). We state the complexity below, using the specific $\alpha$ values from Theorem 3.3 to obtain the displayed per-surrogate budgets.

**Proposition 3.5.** *Fix an outer step $t$ and let $\varepsilon_{t,k} := Q_{\theta_{t,k}} - TQ_{\hat\theta_t}$ denote the error at inner step $k$. Let $\phi(x) := \dfrac{\tanh^2(x)}{\log\cosh(x)}$ and set $x = \beta R_t$. Then the number of iterations sufficient to achieve the $\alpha$ choices suggested by Theorem 3.3 using Eq. 9 satisfies*

$$K_{\mathrm{MSE}} \geq \frac{1}{\underline{w}}\left(\log\frac{1}{\underline{w}} + 2\log\frac{1+\gamma}{1-\gamma}\right),$$

$$K_{\mathrm{LC}} \geq \frac{2}{\underline{w}\,\phi(\beta R_t)}\left(\log\frac{1}{\underline{w}} + \log\frac{1+\gamma}{1-\gamma}\right),$$

$$K_{\mathrm{LSC}} \geq \frac{4}{\underline{p}_t\,\phi(\beta R_t)}\,\log\frac{1+\gamma}{1-\gamma}.$$

*where $p(\varepsilon) := \min_i \dfrac{w_i \cosh(\beta(\varepsilon)_i)}{\sum_j w_j \cosh(\beta(\varepsilon)_j)}$ and $\underline{p}_t := \min_{0 \leq k \leq K} \min_i p_i(\varepsilon_{t,k})$. $\phi(x)$ is decreasing with $\phi(0) = 2$ and as $x \to \infty$, $\phi(x) \approx 1/x$.*

Proposition 3.5 gives a clean decomposition of inner-loop complexity into two conceptually distinct factors. First, the *outer geometry* fixes how much surrogate decrease is needed per target refresh, captured entirely by the $\alpha$-condition through a $\log(1/\alpha)$ requirement. Second, the *inner curvature* determines the *true optimization geometry*; it controls the constant multiplying $\log(1/\alpha)$, i.e., the per-step progress achievable by steepest descent in $\|\cdot\|_\infty$. This separation clarifies how different surrogates behave. For MSE and soft-Huber, both the admissible $\alpha$ and the curvature coefficient inherit an unfavorable dependence on worst-case replay coverage through $\underline{w}$. In contrast, soft-$\ell_\infty$ *decouples the outer requirement from $\underline{w}$* (via the $w$-independent $\alpha$ range in Theorem 3.3) and its inner curvature is *adaptively reweighted* through $p(\cdot)$, concentrating optimization effort on the largest residual coordinates. When $\beta R_t = O(1)$, $\phi(\beta R_t)$ remains a stable constant and the reweighting factor $p(\cdot)$ stays well-behaved on the bounded region that contains the iterates, yielding a more favorable and predictable optimization geometry than MSE/Huber while simultaneously avoiding worst-case coverage constants in the outer loop. This also motivates a practical rule: schedule $\beta$ to keep $\beta R_t \approx O(1)$, preventing curvature degradation and avoiding overly peaked reweighting. Finally, since Adam/AdamW

often behaves like a normalized-sign method in $\ell_\infty$ geometry, the same qualitative dependence of $K$ on $\log(1/\alpha)$ and the curvature coefficients carries over up to constant factors (Balles et al., 2020).

### 3.3. Algorithms

Theorem 3.3 and Proposition 3.5 together motivate an *adaptive* schedule for the temperature factor $\beta$ to meet the required $\alpha$-condition while maintaining good inner-loop conditioning. For soft Huber and soft $\ell_\infty$, this policy lets early iterations use greater effective curvature (small $\beta$), while later iteration increase $\ell_\infty$ alignment and reduce the radius of the asymptotic neighborhood. The resulting adaptive surrogate method is summarized in Algorithm 1.

---

**Algorithm 1** Deep Q-Learning with Adaptive Surrogate

---

**Input:** outer iterations $T$; inner iterations $K_{\mathrm{target}}$; discount $\gamma$; exploration $\epsilon$; stepsize $\eta$; surrogate loss $\mathcal{L}(\cdot;\beta)$ (MSE / soft-Huber / soft-$\ell_\infty$); `use_adaptive_beta` $\in \{0,1\}$; controller $c > 0$; clip range $[\beta_{\min}, \beta_{\max}]$; EMA rate $\rho \in (0,1)$
Initialize parameters $\theta$; set target parameters $\hat\theta \leftarrow \theta$;
Initialize $\beta \leftarrow \beta_{\min}$; $\quad \bar{E} \leftarrow 1$; $\quad$ observe initial state $s$
**for** $t = 1$ **to** $T$ **do**
    **for** $k = 1$ **to** $K_{\mathrm{target}}$ **do**
        Select action $a$ via $\epsilon$-greedy from $Q_\theta(s, \cdot)$
        Execute $a$; store transition $(s, a, r, s', d)$ in $\mathcal{B}$; $s \leftarrow s'$
        **if** $d = 1$ **then**
            Reset environment; observe new initial state $s$
        **end if**
        Sample minibatch $\{(s_i, a_i, r_i, s'_i, d_i)\}_{i=1}^m \sim \mathcal{B}$
        For each $i = 1, \ldots, m$
        Set $Y_i \leftarrow r_i + \gamma(1 - d_i) \max_{a' \in \mathcal{A}} Q_{\hat\theta}(s'_i, a')$
        Set $Q_i \leftarrow Q_\theta(s_i, a_i)$ and $\varepsilon_i \leftarrow Q_i - Y_i$
        **if** `use_adaptive_beta = 1` **then**
            $E \leftarrow \max_{i=1,\ldots,m} |\varepsilon_i|$ (i.e., $E = \|\varepsilon\|_\infty$)
            $\bar{E} \leftarrow (1 - \rho)\bar{E} + \rho E$; $\beta \leftarrow \operatorname{clip}\left(\frac{c}{\bar{E}}, [\beta_{\min}, \beta_{\max}]\right)$
        **end if**
        $\theta \leftarrow \theta - \eta \nabla_\theta \mathcal{L}(\varepsilon; \beta)$
    **end for**
    Synchronize target network: $\hat\theta \leftarrow \theta$
**end for**
**Output:** $Q_\theta$ and greedy policy $\pi(s) = \arg\max_{a \in \mathcal{A}} Q_\theta(s, a)$

---

## 4. Experiment

**Basic hyperparameter setup.** We train each Q-learning agent for 15M frames per Atari game (Towers et al., 2024). Updates begin after 80k steps and occur every 4 environment steps with batch size 32 sampled from a 1M transition uniform replay buffer; the discount is $\gamma = 0.99$. We employ Double-DQN (Hasselt et al., 2016) targets and synchronize the target network every $K_{\mathrm{target}} = 500$ updates with $\tau = 1.0$ (hard copy, no smoothing). Optimization uses AdamW (Loshchilov & Hutter, 2017) with learning rate $5 \times 10^{-5}$ and default PyTorch (Paszke et al., 2019) hyperparameters (betas $0.9, 0.999$; $\epsilon = 10^{-8}$; weight decay $10^{-2}$).

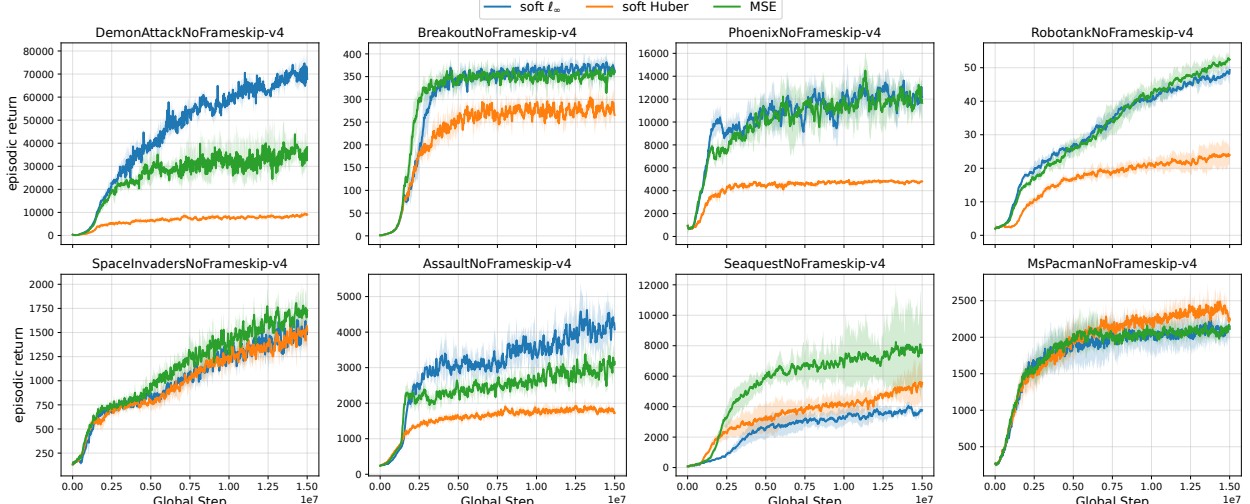

**Figure 1.** Learning curves across Atari games comparing soft $\ell_\infty$ (blue), soft Huber (orange), and MSE (green). *x-axis*: global training steps. *y-axis*: episodic return. Solid lines show the mean over 5 seeds; shaded bands denote the standard deviation.

Exploration follows a linear $\epsilon$-greedy schedule from $1.0$ to $0.01$ over the first $10\%$ of training, after which $\epsilon = 0.01$ is held fixed. Our network follows the standard Atari DQN design of Mnih et al. (2015), comprising a three-layer CNN followed by a 512-unit fully connected layer and a linear action head; full architectural details (kernel sizes, strides, channels) are given in the Appendix. We compare MSE, soft Huber, and soft $\ell_\infty$. For soft Huber and soft $\ell_\infty$ we use an adaptive temperature $\beta_t = \text{clip}\left(\frac{C}{\bar{E}_t}, [\beta_{\min}, \beta_{\max}]\right)$, where $\bar{E}_t$ is an exponential moving average of the 0.95-quantile of error computed on the training minibatch, and $[\beta_{\min}, \beta_{\max}] = [0.1, 20]$. We set $C = 1$ for both soft $\ell_\infty$ and soft Huber. At the beginning of training, network outputs are near zero and $\bar{E}$ can be trivially small. To prevent the large $\beta_t$ at initialization we apply a warm-up: $\beta_t = 0.5$ during the first $20\%$ of training frames, after which the controller above is enabled. Every 50k steps we perform greedy ($\epsilon = 0$) evaluation for 5 episodes in a non-clipped evaluation environment. All curves report mean $\pm$ std over 5 seeds. Our implementation builds on CleanRL (Huang et al., 2022), including its Atari wrappers and replay buffer utilities.

**Experimental results.** Figs. 1–2 summarize our results under the standard setup. We evaluate the three surrogate functions and report (i) episodic return of the greedy policy on evaluation environment and (ii) the maximum Bellman residual $R_\infty$ estimated on a fixed held-out evaluation set $\mathcal{D}_{\text{hold}}$ sampled from the replay buffer: $R_\infty(t) = \max_{(s,a) \in \mathcal{D}_{\text{hold}}} \left| Q_{\theta_t}(s, a) - Y_t(s, a, s') \right|$. In all but one game, the return ordering aligns with the max-error ordering: the method with smaller $R_\infty$ attains a higher final return; when returns tie, the corresponding $R_\infty$ trajectories are nearly indistinguishable over long horizons. Soft $\ell_\infty$

dominates or ties among almost all games except for MsPacman and Seaquest: it achieves the highest or tied-highest return and the smallest $R_\infty$ in most games. MSE is consistently intermediate, while soft Huber exhibits the largest $R_\infty$ and the lowest returns.

The log sum cosh structure underlying soft $\ell_\infty$ induces a soft-max reweighting across samples: its gradient factors as $p_i \tanh(\beta \varepsilon_i)$ with $p_i \propto \cosh(\beta|\varepsilon_i|)$, emphasizing weights on the largest residuals; it therefore targets the maximum error and shrinks $R_\infty$ fastest. MSE allocates gradient mass linearly in $|\varepsilon|$, balancing average and worst-case reduction and typically landing in the middle on both $R_\infty$ and return. Soft Huber is sub-linear in the tail (effectively $\ell_1$-like), down-weighting large residuals; this robustness leaves a larger $R_\infty$ and, consequently, a larger worst-case performance gap that matches the observed ordering. This ordering aligns with the well-established performance bounds (Williams & Baird, 1993): for any state-action value function $Q$ and the policy $\pi$ greedy with respect to $Q$, the actual return from this policy $V^\pi$ satisfies: $\|V^\star - V^\pi\|_\infty \leq \frac{1}{1-\gamma} \|TQ - Q\|_\infty$. Hence, faster reduction of the worst-case Bellman residual yields a uniformly tighter bound on suboptimality, typically correlating with better final return. In addition to the main comparison, we ablate (i) model capacity and (ii) the inner-loop budget, and report these results in Appendix E.

**Effect of inner-loops** We study how the number of inner updates per target refresh affects learning across the surrogates. Agents are trained on `CartPole-v1` for $6 \times 10^5$ environment steps with a 3-layer MLP, AdamW (step size $2.5 \times 10^{-4}$), and otherwise default settings. We vary the inner–loop budget $K \in \{1, 10, 100, 500\}$. In Fig. 3 we observe that increasing $K$ consistently improves rate of con-

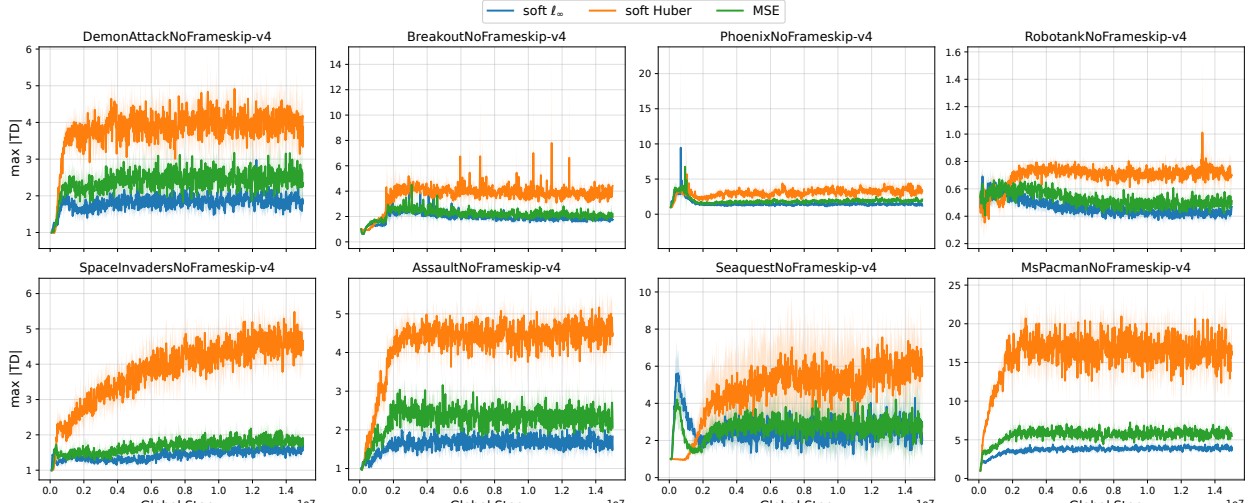

*Figure 2.* Max absolute Bellman error measured on a fixed held-out evaluation set. *x-axis*: global training steps. *y-axis*: max Bellman error $\|Q_{\theta_t} - Y_t\|_\infty$. Solid lines show the mean over 5 seeds; shaded areas denote standard deviation.

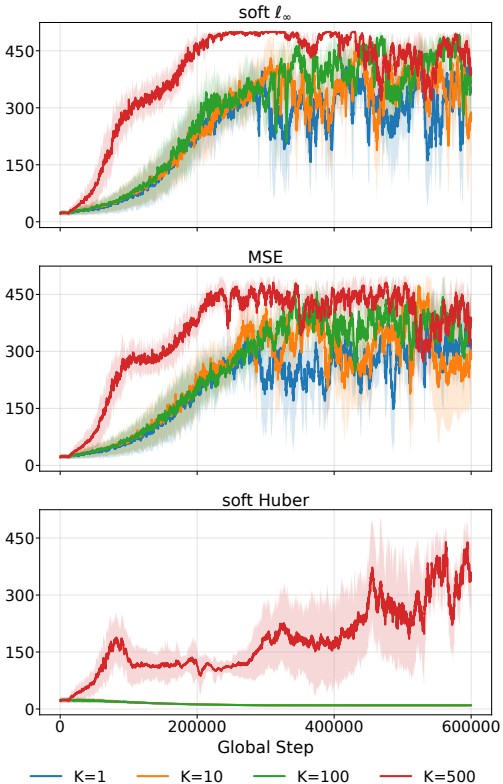

*Figure 3.* Episodic return on CartPole-v1 environment.

vergence and final return across all surrogates. Small $K$ can yield early gains but does not reliably stabilize or saturate at high return. Among all surrogates, soft $\ell_\infty$ climbs fastest and attains the strongest plateau, MSE is intermediate, and soft Huber underweights large residuals and lags unless $K$ is large. These trends are consistent with our theory: more inner–loop reduction produces a better approximation to the target, which translates into stronger contraction in prediction space, yielding faster convergence and saturation.

## 5. Conclusion

We presented a surrogate-optimization view of DQN that makes the geometry mismatch explicit and quantifiable. By extending existing $\alpha$-condition analysis from Euclidean settings to the $\ell_\infty$ geometry, we derived surrogate-specific thresholds that convert inner-loop reduction into $\ell_\infty$ contraction of the Bellman error. This yields convergence guarantees for MSE and (soft) Huber under fixed targets and motivates a temperature-controlled soft-$\ell_\infty$ loss that imposes the weakest admissible $\alpha$-condition while smoothly interpolating between robust quadratic–linear behavior and direct worst-case control.

Empirically, the soft-$\ell_\infty$ surrogate consistently reduces the maximum Bellman residual and matches or exceeds the returns of standard objectives; increasing inner-loop steps reliably improves sample efficiency and final performance, aligning with the theory. Taken together, these results offer a simple, practical recipe for stable value-based control with fixed targets. We note that our guarantees rely on standard coverage and Bellman completeness; relaxing these assumptions and extending the analysis to broader setups would be promising directions for future work.

## Impact Statement

This paper studies the convergence behavior of fixed-target deep Q-learning through a surrogate optimization perspective, with the primary goal of improving the theoretical understanding and practical reliability of value-based reinforcement learning. More stable and predictable Q-learning methods can benefit applications where learning-based control is used, including robotics, operations, and resource management, by reducing sensitivity to hyperparameters and failure modes during training.

At the same time, improvements in reinforcement-learning algorithms may enable stronger autonomy in domains with societal risk, and could be incorporated into systems that cause harm if misused (e.g., unsafe automated decision-making or dual-use applications). Our contribution is methodological and analytical rather than providing new capabilities targeted to a particular downstream task; nonetheless, it may lower the barrier to deploying value-based RL by increasing training stability. We encourage practitioners to pair advances in RL training with standard safety practices, including careful offline evaluation, monitoring, and domain-specific constraints.

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

## A. Connection between Huber and Soft Huber (log cosh)

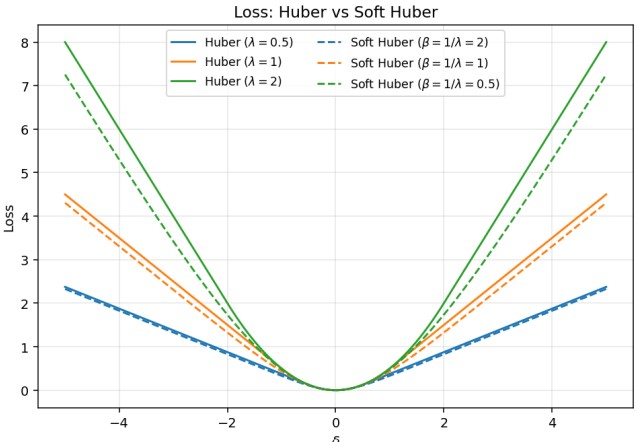

*Figure 4.* Huber vs. soft Huber (log–cosh) losses for matched pairs $(\lambda, \beta = 1/\lambda)$.

**Definitions.** For a residual $u \in \mathbb{R}$ and threshold $\lambda > 0$, the (symmetric) Huber loss is

$$\mathcal{L}_{\text{Huber}}(u; \lambda) = \begin{cases} \frac{1}{2}\, u^2, & |u| \leq \lambda, \\ \lambda\big(|u| - \frac{1}{2}\,\lambda\big), & |u| > \lambda. \end{cases} \tag{10}$$

Its gradient is $\nabla \mathcal{L}_{\text{Huber}}(u; \lambda) = u$ for $|u| \leq \lambda$ and $\lambda \operatorname{sign}(u)$ otherwise.

The *soft Huber* (log cosh) loss we use is the **scaled** form

$$\mathcal{L}_{\text{LC}}(u; \lambda) \;=\; \lambda^2 \, \log \cosh\!\left(\frac{u}{\lambda}\right) \;\equiv\; \frac{1}{\beta^2}\, \log \cosh(\beta\, u), \qquad \beta = \frac{1}{\lambda}. \tag{11}$$

Then $\nabla \mathcal{L}_{\text{LC}}(u; \lambda) = \lambda \tanh\big(u/\lambda\big)$ and $\nabla^2 \mathcal{L}_{\text{LC}}(u; \lambda) = \operatorname{sech}^2\big(u/\lambda\big) \in (0, 1]$.

**Mapping between $\lambda$ and $\beta$.** Eq. equation 11 shows the one-to-one correspondence

$$\boxed{\beta = \tfrac{1}{\lambda}\,, \qquad \mathcal{L}_{\text{LC}}(u; \lambda) = \lambda^2 \log \cosh(u/\lambda) = \tfrac{1}{\beta^2} \log \cosh(\beta u)}$$

$\mathcal{L}_{\text{LC}}$ coincides with $\mathcal{L}_{\text{Huber}}$ up to a additive constant in the tails (the transition at $|u| = \lambda$ is smoothed).

Both losses interpolate between $\ell_2$ (near zero) and $\ell_1$ (for residuals greater than threshold), hence maintain bounded influence. Compared to $L_{\text{Huber}}$, $\mathcal{L}_{\text{LC}}$ is $C^\infty$ (no kink at $|u| = \lambda$), has 1-Lipschitz gradient in the normalized coordinate.

Figure 4 compares the *loss* curves of Huber and soft Huber for three matched pairs $(\lambda, \beta = 1/\lambda)$, and Fig. 5 shows the corresponding *gradients*. The plots illustrate the quadratic core, linear tails with identical slope $\lambda$, and the smooth saturation of $\tanh(\cdot)$.

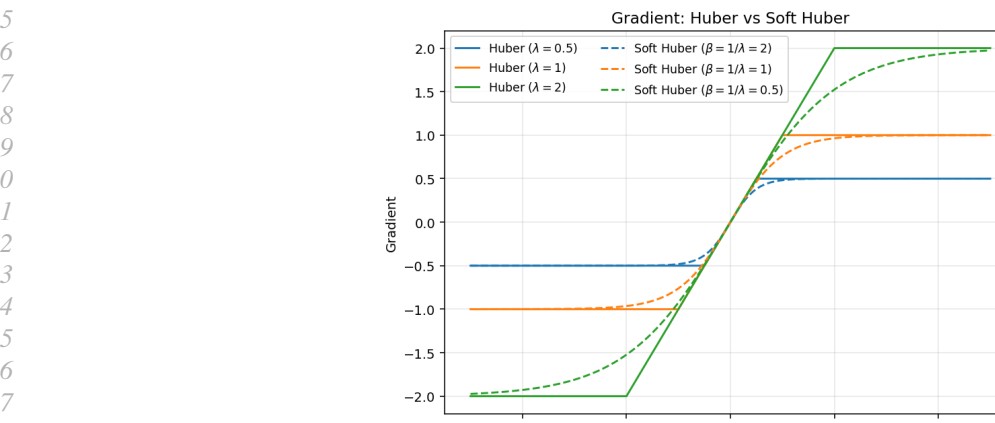

*Figure 5.* Gradients for Huber vs. soft Huber. Soft Huber replaces the hard plateau by the smooth $\tanh$ saturation.

## B. Proof of Theorem 3.3 and Remark 3.4

### B.1. Setup and outer-loop recursion

We work with $Q_t \equiv Q_{\theta_t}$, the Bellman operator $T$, and the outer target

$$Z_t \ = \ (1 - \eta)\, Q_t + \eta\, T Q_t, \qquad \eta \in (0, 1].$$

Let $E_t := \|Q_t - Q^\star\|_\infty$ and let $w$ be a probability vector over $(s, a)$ with $w_{\min} := \min_{s,a} w(s, a) > 0$. The inner loop accepts $\theta_{t+1}$ when

$$\mathcal{L}_{\mathrm{sur}}\big(Q_{t+1} - Z_t, \beta\big) - \mathcal{L}^\star_{\mathrm{sur},t} \ < \ \alpha_{\mathrm{sur}}\big(\mathcal{L}_{\mathrm{sur}}\big(Q_t - Z_t, \beta\big) - \mathcal{L}^\star_{\mathrm{sur},t}\big).$$

for a surrogate loss $\ell_t$ defined below in each case. We will repeatedly use:

$$\|Q_t - T Q_t\|_\infty \ \leq \ \|Q_t - Q^\star\|_\infty + \|T Q_t - T Q^\star\|_\infty \ \leq \ (1 + \gamma)\, E_t,$$

and the standard outer recursion (triangle inequality with $\|Q_{t+1} - Q^\star\|_\infty \leq \|Q_{t+1} - Z_t\|_\infty + \|Z_t - Q^\star\|_\infty$ and $\|Z_t - Q^\star\|_\infty \leq \|(1 - \eta) Q_t + \eta T Q_t - Q^\star\| \leq (1 - \eta) E_t + \eta \gamma E_t \leq (1 - \eta(1 - \gamma)) E_t$):

$$E_{t+1} \ \leq \ \underbrace{(1 - \eta(1 - \gamma))\, E_t}_{\text{error contraction}} + \underbrace{\|Q_{t+1} - Z_t\|_\infty}_{\text{inexactness}}. \tag{12}$$

Assumption 3.2 guarantees that the distribution $w$ places nonzero weight on every state–action pair used by an optimal policy. As a result, any decrease in a weighted surrogate objective necessarily translates into a decrease of the largest Bellman error on the optimal actions. Consequently, once the inner loop achieves the stated $\alpha$-reduction, the Bellman optimality update inherits a strict decrease in $\ell_\infty$ norm, and the outer iteration contracts and converges.

### B.2. Surrogate-to-$\ell_\infty$ comparison lemmas

**Lemma B.1** (Weighted $\ell_2$ vs. $\ell_\infty$). *For any $x$,*

$$\sqrt{w_{\min}}\, \|x\|_\infty \ \leq \ \|x\|_w \ \leq \ \|x\|_\infty.$$

*Proof. (Upper) Since $|x_i| \leq \|x\|_\infty$ and $\sum w = 1$, $\|x\|_w^2 = \sum w\, x^2 \leq \sum w\, \|x\|_\infty^2 = \|x\|_\infty^2$. (Lower) Let $x_i$ achieve $\|x\|_\infty$. Then $\|x\|_w^2 = \sum w\, x^2 \geq w_i \|x\|_\infty^2 \geq w_{\min} \|x\|_\infty^2$. Taking square roots gives the claim.* $\square$

**Lemma B.2** (Soft-Huber vs. $\ell_\infty$). *For any vector $x$,*

$$\frac{w_{\min}}{\beta}\, \|x\|_\infty - \frac{w_{\min}}{\beta^2} \log 2 \ \leq \ \frac{1}{\beta^2} \sum_{s,a} w(s, a)\, \mathrm{logcosh}\big(\beta x(s, a)\big) \ \leq \ \frac{1}{\beta}\, \|x\|_\infty.$$

Proof. *(Upper)* $\log \cosh(u) \leq |u|$ *pointwise* $\Rightarrow \frac{1}{\beta^2} \sum_{s,a} w(s,a) \log \cosh(\beta x(s,a)) \leq \frac{1}{\beta} \sum_{s,a} w(s,a) |x(s,a)| \leq \frac{1}{\beta} \|x\|_\infty$
*since* $\sum w = 1$. *(Lower)* $\sum_{s,a} w(s,a) \log \cosh(\beta x(s,a)) \geq w_{\min} \log \cosh(\beta \|x\|_\infty)$ *and* $\log \cosh(z) \geq z - \log 2$ *for*
$z \geq 0$ *yields* $\frac{1}{\beta^2} \sum_{s,a} w(s,a) \log \cosh(\beta x(s,a)) \geq \frac{w_{\min}}{\beta} \|x\|_\infty - \frac{w_{\min}}{\beta^2} \log 2$. $\qquad \square$

**Lemma B.3** (Soft-$\ell_\infty$ vs. $\ell_\infty$). *For any vector* $x$,

$$\|x\|_\infty + \frac{1}{\beta} \log \frac{w_{\min}}{2} \; \leq \; \frac{1}{\beta} \log \sum_{s,a} w(s,a) \cosh(\beta x(s,a)) \; \leq \; \|x\|_\infty.$$

Proof. *(Upper)* $\cosh(\beta x(s,a)) \leq e^{\beta \|x\|_\infty} \Rightarrow \sum_{s,a} w(s,a) \cosh(\beta x(s,a)) \leq e^{\beta \|x\|_\infty}$, *hence the RHS. (Lower)*
$\sum_{s,a} w(s,a) \cosh(\beta x(s,a)) \geq w_{\min} \cosh(\beta \|x\|_\infty) \geq w_{\min} e^{\beta \|x\|_\infty}/2$, *so the LHS follows by taking* $(1/\beta) \log(\cdot)$. $\qquad \square$

### B.3. Proof of Theorem 3.3

Recall the notation $E_t := \|Q_t - Q^\star\|_\infty$, the error recursion 12 gives, for all $t$,

$$E_{t+1} \; \leq \; \gamma E_t + \|Q_{t+1} - TQ_t\|_\infty. \tag{13}$$

We also repeatedly use

$$\|Q_t - TQ_t\|_\infty \; \leq \; (1+\gamma) E_t, \tag{14}$$

Since each $Q_t \in Z$, the intrinsic approximation radius $d_\infty := \inf_{Q \in Z} \|Q - Q^\star\|_\infty$ yields $d_\infty \leq \liminf_{t \to \infty} E_t$, so it remains to upper bound $\limsup E_t$ for each surrogate. Define $Q_t^\infty := \arg \min_{Q \in Z} \|Q - TQ_t\|_\infty$, then by assumption 3.1 we have that:

$$\|Q_t^\infty - TQ_t\|_\infty \leq \varepsilon_\infty, \quad \forall t$$

**MSE surrogate.** At iteration $t$, the MSE surrogate is

$$\mathcal{L}_{\mathrm{MSE},t}(Q) := \frac{1}{2} \sum_x w(x) \big(Q(x) - TQ_t(x)\big)^2.$$

Lemma B.1 gives, for any $Q$,

$$\frac{w_{\min}}{2} \|Q - TQ_t\|_\infty^2 \; \leq \; \mathcal{L}_{\mathrm{MSE},t}(Q) \; \leq \; \frac{1}{2} \|Q - TQ_t\|_\infty^2. \tag{15}$$

Let $Q_t^\star$ be a minimizer of $\mathcal{L}_{\mathrm{MSE},t}$ over $Z$. Then

$$\mathcal{L}_{\mathrm{MSE},t}(Q_t^\star) \; \leq \; \mathcal{L}_{\mathrm{MSE},t}(Q_t^\infty) \; \leq \; \tfrac{1}{2} \|Q_t^\infty - TQ_t\|_\infty^2 \; \leq \; \tfrac{1}{2} \varepsilon_\infty^2.$$

The $\alpha$-condition equation 8, together with equation 15, implies

$$\frac{w_{\min}}{2} \|Q_{t+1} - TQ_t\|_\infty^2 \leq \mathcal{L}_{\mathrm{MSE},t}(Q_{t+1})$$
$$\leq \alpha \, \mathcal{L}_{\mathrm{MSE},t}(Q_t) + (1-\alpha) \, \mathcal{L}_{\mathrm{MSE},t}(Q_t^\star)$$
$$\leq \frac{1}{2} \Big[ \alpha \|Q_t - TQ_t\|_\infty^2 + (1-\alpha)\varepsilon_\infty^2 \Big],$$

so

$$\|Q_{t+1} - TQ_t\|_\infty^2 \; \leq \; \frac{\alpha}{w_{\min}} \|Q_t - TQ_t\|_\infty^2 + \frac{1-\alpha}{w_{\min}} \varepsilon_\infty^2.$$

Taking square roots and using $\sqrt{u+v} \leq \sqrt{u} + \sqrt{v}$,

$$\|Q_{t+1} - TQ_t\|_\infty \; \leq \; \sqrt{\frac{\alpha}{w_{\min}}} \|Q_t - TQ_t\|_\infty + \sqrt{\frac{1-\alpha}{w_{\min}}} \varepsilon_\infty.$$

Applying equation 14, $\|Q_t - TQ_t\|_\infty \leq (1+\gamma)E_t$, we obtain

$$\|Q_{t+1} - TQ_t\|_\infty \ \leq \ (1+\gamma)\sqrt{\frac{\alpha}{w_{\min}}}\, E_t + \sqrt{\frac{1-\alpha}{w_{\min}}}\, \varepsilon_\infty.$$

Substituting this into equation 13 yields

$$E_{t+1} \ \leq \ \left[\gamma + (1+\gamma)\sqrt{\frac{\alpha}{w_{\min}}}\right] E_t + \sqrt{\frac{1-\alpha}{w_{\min}}}\, \varepsilon_\infty.$$

When $\alpha_{\mathrm{MSE}} < w_{\min}\left(\frac{1-\gamma}{1+\gamma}\right)^2$, i.e., $\gamma + (1+\gamma)\sqrt{\alpha/w_{\min}} < 1$, the linear recursion gives

$$\limsup_{t\to\infty} E_t \ \leq \ \frac{\sqrt{\frac{1-\alpha}{w_{\min}}}\, \varepsilon_\infty}{1 - \gamma - (1+\gamma)\sqrt{\frac{\alpha}{w_{\min}}}}.$$

**Soft Huber surrogate** (LC).    Following Lemma B.2 we have that, for all $Q$,

$$\frac{w_{\min}}{\beta}\|Q - TQ_t\|_\infty - \frac{w_{\min}}{\beta^2}\log 2 \ \leq \ \mathcal{L}_{\mathrm{LC},t}(Q) \ \leq \ \frac{1}{\beta}\|Q - TQ_t\|_\infty. \tag{16}$$

Let $Q_t^\star$ minimize $\mathcal{L}_{\mathrm{LC},t}$ over $Z$, then

$$\mathcal{L}_{\mathrm{LC},t}(Q_t^\star) \ \leq \ \mathcal{L}_{\mathrm{LC},t}(Q_t^\infty) \ \leq \ \frac{1}{\beta}\, \varepsilon_\infty.$$

Using the $\alpha$-condition equation 8 and equation 16,

$$\frac{w_{\min}}{\beta}\|Q_{t+1} - TQ_t\|_\infty - \frac{w_{\min}}{\beta^2}\log 2 \leq \mathcal{L}_{\mathrm{LC},t}(Q_{t+1})$$
$$\leq \alpha\,\mathcal{L}_{\mathrm{LC},t}(Q_t) + (1-\alpha)\,\mathcal{L}_{\mathrm{LC},t}(Q_t^\star)$$
$$\leq \alpha \cdot \frac{1}{\beta}\|Q_t - TQ_t\|_\infty + (1-\alpha)\cdot\frac{1}{\beta}\,\varepsilon_\infty,$$

so after multiplying by $\beta$,

$$w_{\min}\|Q_{t+1} - TQ_t\|_\infty \ \leq \ \alpha\,\|Q_t - TQ_t\|_\infty + (1-\alpha)\,\varepsilon_\infty + \frac{w_{\min}}{\beta}\log 2.$$

Using equation 14,

$$w_{\min}\|Q_{t+1} - TQ_t\|_\infty \ \leq \ \alpha(1+\gamma)E_t + (1-\alpha)\,\varepsilon_\infty + \frac{w_{\min}}{\beta}\log 2.$$

Substituting into equation 13,

$$E_{t+1} \ \leq \ \left[\gamma + \frac{\alpha}{w_{\min}}(1+\gamma)\right]E_t + \frac{(1-\alpha)}{w_{\min}}\,\varepsilon_\infty + \frac{\log 2}{\beta}.$$

If $\alpha < w_{\min}\frac{1-\gamma}{1+\gamma}$, i.e.,the sequence $E_t$ contracts as $t\to\infty$, we have

$$\limsup_{t\to\infty} E_t \ \leq \ \frac{\frac{1-\alpha}{w_{\min}}\,\varepsilon_\infty + \frac{1}{\beta}\log 2}{1 - \gamma - \frac{\alpha}{w_{\min}}(1+\gamma)}.$$

For $\alpha = 0$, we will have

$$\limsup_{t\to\infty} E_t \ \leq \ \frac{\frac{1}{w_{\min}}\,\varepsilon_\infty + \frac{1}{\beta}\log 2}{1 - \gamma},$$

**Soft $\ell_\infty$ surrogate** (LSC).    For the soft-$\ell_\infty$ surrogate Lemma B.3 provides a constant $C_\beta = \frac{1}{\beta} \log \frac{2}{w_{\min}}$ such that, for all $Q$,

$$\|Q - TQ_t\|_\infty - C_\beta \;\leq\; \mathcal{L}_{\mathrm{LSC},t}(Q) \;\leq\; \|Q - TQ_t\|_\infty. \tag{17}$$

Let $Q_t^\star$ minimize $\mathcal{L}_{\mathrm{LSC},t}$ over $Z$, and let $Q_t^\infty \in Z$ satisfy $\|Q_t^\infty - TQ_t\|_\infty \leq \varepsilon_\infty$ (approximate Bellman completeness). Then

$$\mathcal{L}_{\mathrm{LSC},t}(Q_t^\star) \;\leq\; \mathcal{L}_{\mathrm{LSC},t}(Q_t^\infty) \;\leq\; \varepsilon_\infty.$$

Using the $\alpha$-condition equation 8 and equation 17,

$$\begin{aligned}
\|Q_{t+1} - TQ_t\|_\infty - C_\beta &\leq \mathcal{L}_{\mathrm{LSC},t}(Q_{t+1}) \\
&\leq \alpha\,\mathcal{L}_{\mathrm{LSC},t}(Q_t) + (1-\alpha)\,\mathcal{L}_{\mathrm{LSC},t}(Q_t^\star) \\
&\leq \alpha\,\|Q_t - TQ_t\|_\infty + (1-\alpha)\varepsilon_\infty,
\end{aligned}$$

hence, after multiplying by $\beta$,

$$\|Q_{t+1} - TQ_t\|_\infty \;\leq\; \alpha\,\|Q_t - TQ_t\|_\infty + (1-\alpha)\,\varepsilon_\infty + C_\beta. \tag{18}$$

Using equation 14, $\|Q_t - TQ_t\|_\infty \leq (1+\gamma)E_t$ with $E_t := \|Q_t - Q^\star\|_\infty$, we obtain

$$\|Q_{t+1} - TQ_t\|_\infty \;\leq\; \alpha(1+\gamma)E_t + (1-\alpha)\,\varepsilon_\infty + C_\beta. \tag{19}$$

Plugging equation 19 into the error–propagation inequality equation 13,

$$E_{t+1} \;\leq\; \gamma E_t + \|Q_{t+1} - TQ_t\|_\infty,$$

gives

$$E_{t+1} \;\leq\; \big[\gamma + \alpha(1+\gamma)\big]E_t + (1-\alpha)\,\varepsilon_\infty + C_\beta. \tag{20}$$

If

$$\gamma + \alpha(1+\gamma) < 1 \qquad \Longleftrightarrow \qquad \alpha < \frac{1-\gamma}{1+\gamma},$$

the linear recursion equation 20 yields

$$\limsup_{t\to\infty} E_t \;\leq\; \frac{(1-\alpha)\,\varepsilon_\infty + C_\beta}{1 - \gamma - \alpha(1+\gamma)}.$$

For $\alpha = 0$, equation 20 reduces to $E_{t+1} \leq \gamma E_t + \varepsilon_\infty + \beta C_\beta$, so

$$\limsup_{t\to\infty} E_t \;\leq\; \frac{\varepsilon_\infty + C_\beta}{1 - \gamma}.$$

(up to absorbing $\beta$ into $C_\beta$ as in the main text).

This completes the proof of Theorem 3.3.

**B.4. Proof for Remark 3.4**

**Lemma B.4** (Soft Huber vs. $\ell_\infty$). *For any vector $x$,*

$$\frac{w_{\min}}{\beta^2} \log \cosh \beta\|x\|_\infty \;\leq\; \frac{1}{\beta^2} \sum_{s,a} w(s,a)\,\mathrm{logcosh}\big(\beta x(s,a)\big) \;\leq\; \frac{1}{\beta^2} \log \cosh \beta\|x\|_\infty.$$

Proof.    $\frac{1}{\beta^2} \sum_{s,a} w(s,a)\,\mathrm{logcosh}\big(\beta x(s,a)\big) \;\leq\; \frac{1}{\beta^2} \sum_{s,a} w(s,a) \log \cosh(\beta\|x\|_\infty) \;=\; \frac{1}{\beta^2} \log \cosh(\beta\|x\|_\infty)$ *(Lower)*
$\sum_{s,a} w(s,a) \log \cosh(\beta x(s,a)) \geq w_{\min} \log \cosh(\beta\|x\|_\infty)$    $\square$

Then we have, by Lemma B.4,

$$\frac{w_{\min}}{\beta^2} \log \cosh\big(\beta\|Q_{t+1} - Z_t\|_\infty\big) \;\leq\; \mathcal{L}_{\mathrm{LC},t}(\theta_{t+1})$$

and

$$\mathcal{L}_{\text{LC},t}(\theta_t) \leq \frac{1}{\beta^2} \log \cosh\big(\beta\|Q_t - Z_t\|_\infty\big).$$

Now apply the $\alpha$-condition (Eq. 8) including the approximation term:

$$\mathcal{L}_{\text{LC},t}(\theta_{t+1}) \leq \alpha_{\text{LC}}\, \mathcal{L}_{\text{LC},t}(\theta_t) + (1 - \alpha_{\text{LC}})\, \mathcal{L}^\star_{\text{LC},t}. \tag{21}$$

To bound $\mathcal{L}^\star_{\text{LC},t}$ using $\varepsilon_\infty$, by Definition 3.1 there exists $\tilde{Q}_t \in \mathcal{Z}$ such that $\|\tilde{Q}_t - Z_t\|_\infty \leq \varepsilon_\infty$, hence

$$\mathcal{L}^\star_{\text{LC},t} = \min_{Q \in \mathcal{Z}} \mathcal{L}_{\text{LC}}(Q - Z_t) \leq \mathcal{L}_{\text{LC}}(\tilde{Q}_t - Z_t) \leq \frac{1}{\beta^2} \log \cosh(\beta\varepsilon_\infty),$$

where the last inequality uses Lemma B.4 (upper bound).

Combining the above displays yields

$$\frac{w_{\min}}{\beta^2} \log \cosh\big(\beta\|Q_{t+1} - Z_t\|_\infty\big) \leq \frac{\alpha_{\text{LC}}}{\beta^2} \log \cosh\big(\beta\|Q_t - Z_t\|_\infty\big) + \frac{1 - \alpha_{\text{LC}}}{\beta^2} \log \cosh(\beta\varepsilon_\infty). \tag{22}$$

Now let $r_t := \|Q_t - Z_t\|_\infty$ and define $u_t := \log \cosh(\beta r_t)$. Then equation 22 becomes

$$u_{t+1} \leq \frac{\alpha_{\text{LC}}}{w_{\min}} u_t + \frac{1 - \alpha_{\text{LC}}}{w_{\min}} \log \cosh(\beta\varepsilon_\infty).$$

Assuming $\alpha_{\text{LC}} < w_{\min}$, this is a stable affine recursion, so $(u_t)$ is bounded and

$$\limsup_{t\to\infty} u_t \leq C_{\alpha,w}\, \log \cosh(\beta\varepsilon_\infty) \quad \text{for some constant } C_{\alpha,w} < \infty \text{ depending only on } \alpha_{\text{LC}}, w_{\min}.$$

Since $\log \cosh$ is increasing on $[0, \infty)$, this implies

$$\limsup_{t\to\infty} r_t \leq (\log \cosh)^{-1}\Big(C_{\alpha,w}\, \log \cosh(\beta\varepsilon_\infty)\Big)/\beta.$$

Finally, because $\log \cosh(0) = 0$ and $(\log \cosh)^{-1}$ is continuous at 0, the right-hand side goes to 0 as $\varepsilon_\infty \to 0$. In particular, the steady-state residual scales with $\varepsilon_\infty$ and vanishes under Bellman completeness.

Further more we can show in this case we have the same $\alpha$-condition as in MSE: ignoring the approximation term in Eq. 8 (which we already see only contributes an additive steady-state floor handled separately), the multiplicative part of the $\alpha$-condition gives:

$$\frac{w_{\min}}{\beta^2} \log \cosh \beta\|Q_{t+1} - Z_t\|_\infty \leq \mathcal{L}_{\text{LC},t}(\theta_{t+1}) \leq \alpha_{\text{LC}}\, \mathcal{L}_{\text{LC},t}(\theta_t) \leq \frac{\alpha_{\text{LC}}}{\beta^2} \log \cosh \beta\|Q_t - Z_t\|_\infty \tag{23}$$

Now, using the fact that for $a > 1$, and $x > 0$ we have $a \log \cosh(x) \geq \log \cosh(\sqrt{a}x)$, we have that for $\alpha_{\text{LC}} \leq w_{\min}$,

$$\log \cosh \sqrt{\frac{w_{\min}\beta^2}{\alpha_{\text{LC}}}}\|Q_{t+1} - Z_t\|_\infty \leq \frac{w_{\min}}{\alpha_{\text{LC}}} \log \cosh \beta\|Q_{t+1} - Z_t\|_\infty \leq \log \cosh \beta\|Q_t - Z_t\|_\infty \tag{24}$$

and then because the function $\log \cosh^{-1}$ is increasing we have $\sqrt{\frac{w_{\min}}{\alpha_{\text{LC}}}}\|Q_{t+1} - Z_t\|_\infty \leq \|Q_t - Z_t\|_\infty$ which leads to

$$\|Q_{t+1} - Z_t\|_\infty \leq \sqrt{\frac{\alpha_{\text{LC}}}{w_{\min}}}\|Q_t - Z_t\|_\infty \leq \sqrt{\frac{\alpha_{\text{LC}}(1 + \gamma)^2}{w_{\min}}} E_t \tag{25}$$

which leads to

$$E_{t+1} \leq \left[ 1 - \eta(1 - \gamma) + \eta(1 + \gamma)\sqrt{\frac{\alpha_{\text{LC}}}{w_{\min}}} \right] E_t \tag{26}$$

Thus we get the slower but unbiased $\alpha$ condition: $\alpha_{\text{LC}} \leq w_{\min}\big(\frac{1-\gamma}{1+\gamma}\big)^2$.

**Lemma B.5** (Soft-$\ell_\infty$ vs. $\ell_\infty$)**.** *For any vector $x$,*

$$\frac{w_{\min}}{\beta} \log \cosh(\beta \|x\|_\infty) \leq \frac{1}{\beta} \log \sum_{s,a} w(s,a) \cosh(\beta x(s,a)) \leq \frac{1}{\beta} \log \cosh(\beta \|x\|_\infty)$$

Proof. *(Upper)* $\frac{1}{\beta} \log \sum_{s,a} w(s,a) \cosh(\beta x(s,a)) \leq \frac{1}{\beta} \log \sum_{s,a} w(s,a) \cosh(\beta \|x\|_\infty) \leq \frac{1}{\beta} \log \cosh(\beta \|x\|_\infty)$. *(Lower) By concavity of the* log, $\frac{1}{\beta} \log \sum_{s,a} w(s,a) \cosh(\beta x(s,a)) \geq \frac{1}{\beta} \sum_{s,a} w(s,a) \log \cosh(\beta x(s,a)) \geq \frac{w_{\min}}{\beta} \log \cosh(\beta \|x\|_\infty)$. □

Thus similarly as for the soft Huber we get

$$E_{t+1} \ \leq \ \left[ 1 - \eta(1-\gamma) \ + \ (1+\gamma)\,\eta \,\sqrt{\frac{\alpha}{w_{\min}}} \right] E_t \tag{27}$$

and the corresponding slower but unbiased $\alpha$ condition: $\alpha_{\mathrm{LSC}} \leq w_{\min}(\frac{1-\gamma}{1+\gamma})^2$.

## C. Proof for Proposition 3.5

**Basic setup.** We study the inner optimization over the residual vector $\delta = z - Y \in \mathbb{R}^m$, and let $R = \|\delta\|_\infty$; we are using the same network across all surrogates. The geometry we considered here is the $\ell_\infty$ norm, whose dual is $\ell_1$. For a matrix $H$, we use the induced operator norm

$$\|H\|_{\infty,1} \ := \ \max_{\|u\|_\infty \leq 1} \ \|Hu\|_1.$$

Throught the proof we consider the steepest-descent update Eq(9). We will bound smoothness by $L_\infty$ for each surrogates (Step 1), invoke the descent lemma in this geometry (Step 2), use sublevel-set invariance to remain in $\mathcal{B}_\infty(R)$ (Step 3), and establish an $\ell_1$–PL inequality on $\mathcal{B}_\infty(R)$ (Step 4), which together yield the claimed $K$ (Step 5).

**Step 1: $\ell_\infty$-smoothness constants for the surrogates**

**Lemma C.1** ($\ell_\infty$-smoothness)**.** *Let $\delta \in \mathbb{R}^m$ index $(s,a)$ and let $w_i > 0$ with $\sum_i w_i = 1$. Define*

$$\mathcal{L}_{\mathrm{MSE}}(\delta) \ = \ \tfrac{1}{2} \sum_{i=1}^m w_i \delta_i^2, \qquad \mathcal{L}_{\mathrm{LC}}(\delta) \ = \ \tfrac{1}{\beta^2} \sum_{i=1}^m w_i \log \cosh(\beta \delta_i), \qquad \mathcal{L}_{\mathrm{LSC}}(\delta) \ = \ \tfrac{1}{\beta} \log \Big( \sum_{i=1}^m w_i \cosh(\beta \delta_i) \Big).$$

*Then each surrogate is $\ell_\infty$-smooth with the following constants:*

$$\boxed{L_\infty^{\mathrm{MSE}} = 1, \qquad L_\infty^{\mathrm{LC}} \leq 1, \qquad L_\infty^{\mathrm{LSC}} \leq 2\beta.}$$

*Proof.* By Theorem 5.12 in Beck (2017), for twice continuously differentiable $S$, $L$-smoothness w.r.t. a norm $\|\cdot\|$ is equivalent to the uniform Hessian bound $\sup_\delta \|\nabla^2 S(\delta)\|_{\|\cdot\| \to \|\cdot\|_*} \leq L$, where here $\|\cdot\| = \|\cdot\|_\infty$, $\|\cdot\|_* = \|\cdot\|_1$, and the induced operator norm is $\|\cdot\|_{\infty,1}$.

**(a)** $\mathcal{L}_{\mathrm{MSE}}$**.** $\nabla^2 \mathcal{L}_{\mathrm{MSE}}(\delta) = \mathrm{diag}(w)$ for all $\delta$. For any $u$ with $\|u\|_\infty \leq 1$,

$$\big\| \nabla^2 \mathcal{L}_{\mathrm{MSE}}\, u \big\|_1 = \sum_{i=1}^m w_i |u_i| \ \leq \ \sum_{i=1}^m w_i \ = \ 1.$$

Taking the supremum over $\|u\|_\infty \leq 1$ yields $\|\nabla^2 \mathcal{L}_{\mathrm{MSE}}\|_{\infty,1} = 2$.

**(b)** $\mathcal{L}_{\mathrm{LC}}$**.** $\nabla^2 \mathcal{L}_{\mathrm{LC}}(\delta) = \mathrm{diag}\big(w_i \operatorname{sech}^2(\beta \delta_i)\big)$ for all $\delta$. Hence, for any $\|u\|_\infty \leq 1$,

$$\big\| \nabla^2 \mathcal{L}_{\mathrm{LC}}\, u \big\|_1 = \sum_{i=1}^m w_i \operatorname{sech}^2(\beta \delta_i)\, |u_i| \ \leq \ \sum_{i=1}^m w_i \operatorname{sech}^2(\beta \delta_i) \ \leq \ \sum_{i=1}^m w_i \ = \ 1,$$

since $|u_i| \leq 1$ and $\operatorname{sech}^2(\cdot) \leq 1$. Thus $\|\nabla^2 \mathcal{L}_{\mathrm{LC}}\|_{\infty,1} \leq 1$ globally.

**(c) $\mathcal{L}_{\mathrm{LSC}}$.** Let $\pi_i := \dfrac{w_i \cosh(\beta \delta_i)}{\sum_{j=1}^m w_j \cosh(\beta \delta_j)}$, $t_i := \tanh(\beta \delta_i)$, and $a_i := \pi_i t_i$. Then $\nabla \mathcal{L}_{\mathrm{LSC}}(\delta) = a$ and a direct differentiation gives the Hessian $\nabla^2 \mathcal{L}_{\mathrm{LSC}}(\delta) = \beta\big(\mathrm{diag}(\pi) - aa^\top\big)$ for all $\delta$. For any $u$ with $\|u\|_\infty \leq 1$,

$$\left\|\nabla^2 \mathcal{L}_{\mathrm{LSC}}\, u\right\|_1 = \beta \left\|\mathrm{diag}(\pi)u - a(a^\top u)\right\|_1 \;\leq\; \beta\Big(\left\|\mathrm{diag}(\pi)u\right\|_1 + \|a\|_1\,|a^\top u|\Big).$$

Since $\sum_i \pi_i = 1$ and $\|u\|_\infty \leq 1$, we have $\|\mathrm{diag}(\pi)u\|_1 = \sum_i \pi_i|u_i| \leq 1$. Also, $\|a\|_1 = \sum_i \pi_i|t_i| \leq \sum_i \pi_i = 1$, and $|a^\top u| \leq \|a\|_1 \|u\|_\infty \leq 1$. Consequently,

$$\left\|\nabla^2 \mathcal{L}_{\mathrm{LSC}}\, u\right\|_1 \;\leq\; \beta(1+1) \;=\; 2\beta,$$

uniformly over $\delta$ and $u$, hence $\|\nabla^2 \mathcal{L}_{\mathrm{LSC}}\|_{\infty,1} \leq 2\beta$. $\qquad\square$

**Step 2: Descent in $\ell_\infty$ geometry.** We apply the standard descent inequality for the steepest-descent step (Eq. equation 9) in the $\ell_\infty$ geometry.

**Lemma C.2** (Descent lemma in $\ell_\infty$ geometry (Lemma 2 of Balles et al. (2020))). *If a differentiable function $\mathcal{L}$ is $L_\infty$-smooth (i.e., $\|\nabla^2 \mathcal{L}\|_{\infty,1} \leq L_\infty$), then the update*

$$\delta_{k+1} = \delta_k - \frac{\|\nabla \mathcal{L}(\delta_k)\|_1}{L_\infty}\, \mathrm{sign}\big(\nabla \mathcal{L}(\delta_k)\big)$$

*satisfies*

$$\mathcal{L}(\delta_{k+1}) \;\leq\; \mathcal{L}(\delta_k) \;-\; \frac{1}{2L_\infty}\,\|\nabla \mathcal{L}(\delta_k)\|_1^2.$$

Specializing Lemma C.2 with the smoothness constants from Lemma C.1 gives

$$\text{MSE:}\quad \mathcal{L}_{\mathrm{MSE}}(\delta_{k+1}) \;\leq\; \mathcal{L}_{\mathrm{MSE}}(\delta_k) \;-\; \tfrac{1}{2}\left\|\nabla \mathcal{L}_{\mathrm{MSE}}(\delta_k)\right\|_1^2,$$

$$\text{LC:}\quad \mathcal{L}_{\mathrm{LC}}(\delta_{k+1}) \;\leq\; \mathcal{L}_{\mathrm{LC}}(\delta_k) \;-\; \tfrac{1}{2}\left\|\nabla \mathcal{L}_{\mathrm{LC}}(\delta_k)\right\|_1^2,$$

$$\text{LSC:}\quad \mathcal{L}_{\mathrm{LSC}}(\delta_{k+1}) \;\leq\; \mathcal{L}_{\mathrm{LSC}}(\delta_k) \;-\; \tfrac{1}{4\beta}\left\|\nabla \mathcal{L}_{\mathrm{LSC}}(\delta_k)\right\|_1^2.$$

**Step 3: Coercivity $\Rightarrow$ bounded $\|\delta\|_\infty$ (region invariance)** We show the surrogates are $\ell_\infty$-*coercive*, i.e., $\mathcal{L}(\delta) \to \infty$ whenever $\|\delta\|_\infty \to \infty$. This implies all sublevel sets $\{\mathcal{L} \leq S_0\}$ are bounded in $\|\cdot\|_\infty$, so iterates produced by the descent step equation 9 remain in a compact region where the smoothness bounds of Step 1 apply.

**Definition C.3** ($\ell_\infty$-coercivity). A function $\mathcal{L}: \mathbb{R}^m \to \mathbb{R}$ is $\ell_\infty$-*coercive* if $\mathcal{L}(\delta) \to \infty$ whenever $\|\delta\|_\infty \to \infty$.

**Lemma C.4** (Coercivity of $\mathcal{L}_{\mathrm{MSE}}$, $\mathcal{L}_{\mathrm{LC}}$, and $\mathcal{L}_{\mathrm{LSC}}$). *Let $R := \|\delta\|_\infty$ and pick $i^\star$ such that $|\delta_{i^\star}| = E$. With $w_{\min} := \min_i w_i > 0$,*

$$\text{MSE:}\quad \mathcal{L}_{\mathrm{MSE}}(\delta) = \tfrac{1}{2}\sum_i w_i\, \delta_i^2 \;\geq\; \tfrac{w_{\min}}{2}\, R^2 \;\xrightarrow[R\to\infty]{}\; \infty.$$

$$\text{LC:}\quad \mathcal{L}_{\mathrm{LC}}(\delta) = \tfrac{1}{\beta^2}\sum_i w_i\, \log\cosh(\beta\delta_i) \;\geq\; \tfrac{w_{\min}}{\beta^2}\, \log\cosh(\beta R) \;\xrightarrow[R\to\infty]{}\; \infty.$$

$$\text{LSC:}\quad \mathcal{L}_{\mathrm{LSC}}(\delta) = \tfrac{1}{\beta} \log\Big(\sum_i w_i\, \cosh(\beta\delta_i)\Big) \;\geq\; \tfrac{1}{\beta}\log\big(w_{\min}\cosh(\beta R)\big) \;\xrightarrow[R\to\infty]{}\; \infty.$$

*Thus all three surrogates are $\ell_\infty$-coercive.*

Since coercivity means $\mathcal{L}(\delta) \to \infty$ whenever $\|\delta\|_\infty \to \infty$, no sublevel set $\{\delta : \mathcal{L}(\delta) \leq S_0\}$ can "reach infinity" in $\|\cdot\|_\infty$ (else it would contain a sequence with $\mathcal{L} \to \infty$), so it is bounded; monotone descent ($\mathcal{L}(\delta_{k+1}) \leq \mathcal{L}(\delta_k)$) then keeps all iterates inside this bounded set:

**Corollary C.5** (Sublevel-set radius bounds). *Fix $S_0 > 0$ and suppose $\mathcal{L}(\delta) \leq S_0$. Then the maximal coordinate $E = \|\delta\|_\infty$ is bounded as*

$$\text{MSE:} \quad \mathcal{L}_{\text{MSE}}(\delta) \leq S_0 \;\Rightarrow\; R \;\leq\; \sqrt{\frac{2S_0}{w_{\min}}}.$$

$$\text{LC:} \quad \mathcal{L}_{\text{LC}}(\delta) \leq S_0 \;\Rightarrow\; \sum_i w_i \log\cosh(\beta\delta_i) \;\leq\; \beta^2 S_0 \;\Rightarrow\; \log\cosh(\beta R) \;\leq\; \frac{\beta^2 S_0}{w_{\min}}$$

$$\Rightarrow\; R \;\leq\; \tfrac{1}{\beta}\operatorname{arcosh}\!\Big(e^{\beta^2 S_0/w_{\min}}\Big).$$

$$\text{LSC:} \quad \mathcal{L}_{\text{LSC}}(\delta) \leq S_0 \;\Rightarrow\; \sum_i w_i \cosh(\beta|\delta_i|) \;\leq\; e^{\beta S_0}$$

$$\Rightarrow\; \cosh(\beta R) \;\leq\; \frac{e^{\beta S_0}}{w_{\min}} \;\Rightarrow\; R \;\leq\; \tfrac{1}{\beta}\operatorname{arcosh}\!\Big(\frac{e^{\beta S_0}}{w_{\min}}\Big).$$

Starting from any $\delta_0$ with finite $\mathcal{L}(\delta_0)$, the descent step equation 9 generates a nonincreasing sequence $\mathcal{L}(\delta_k) \leq \mathcal{L}(\delta_0)$; hence $\delta_k$ remains in the bounded sublevel set of $S_0 := \mathcal{L}(\delta_0)$. Therefore the region $\{\|\delta\|_\infty \leq R\}$ with $R$ given by Corollary C.5 is *invariant* for the inner loop, and the smoothness constants from Lemma C.1 apply throughout.

**Step 4: $\ell_1$–PL inequalities on the population sublevel set.** Fix $S_0 > 0$ and let $R$ be any number such that the sublevel set $\{\delta : \mathcal{L}(\delta) \leq S_0\}$ satisfies $\|\delta\|_\infty \leq R$ (see Corollary C.5). Define

$$\phi(a) \;:=\; \frac{\tanh^2(a)}{\log\cosh(a)}, \qquad a \geq 0,$$

which is decreasing, with $\phi(0) = 2$ and $\phi(a) \sim 1/a$ as $a \to \infty$.

**Lemma C.6** ($\ell_1$–PL on $\{\mathcal{L} \leq S_0\}$). *With population weights $w_i > 0$, $\sum_i w_i = 1$, and $w_{\min} := \min_i w_i$, the following inequalities hold for all $\delta$ with $\|\delta\|_\infty \leq R$:*

$$\boxed{\begin{aligned}
&\frac{1}{2}\left\|\nabla\mathcal{L}_{\text{MSE}}(\delta)\right\|_1^2 \;\geq\; w_{\min}\,\mathcal{L}_{\text{MSE}}(\delta), \qquad \frac{1}{2}\left\|\nabla\mathcal{L}_{\text{LC}}(\delta)\right\|_1^2 \;\geq\; \frac{w_{\min}}{2}\,\phi(\beta R)\,\mathcal{L}_{\text{LC}}(\delta), \\
&\frac{1}{2}\left\|\nabla\mathcal{L}_{\text{LSC}}(\delta)\right\|_1^2 \;\geq\; \frac{\beta}{2}\,p_{\min}(\delta)\,\phi(\beta R)\,\mathcal{L}_{\text{LSC}}(\delta).
\end{aligned}}$$

*Here $p_i(\delta) := \dfrac{w_i\cosh(\beta\delta_i)}{\sum_j w_j\cosh(\beta\delta_j)}$ and $p_{\min}(\delta) := \min_i p_i(\delta)$.*

*Proof. MSE.* $\mathcal{L}_{\text{MSE}}(\delta) = \frac{1}{2}\sum_i w_i\delta_i^2$, so $\nabla\mathcal{L}_{\text{MSE}} = (w_i\delta_i)_i$. Let $x_i := \sqrt{w_i}\,\delta_i$. Then $\|\nabla\mathcal{L}_{\text{MSE}}\|_1 = \sum_i \sqrt{w_i}|x_i| \geq \sqrt{w_{\min}}\|x\|_2$, hence $\|\nabla\mathcal{L}_{\text{MSE}}\|_1^2 \geq 2w_{\min}\mathcal{L}_{\text{MSE}}$.

*LC.* For $\mathcal{L}_{\text{LC}}(\delta) = \beta^{-2}\sum_i w_i\log\cosh(\beta\delta_i)$,

$$\partial_i\mathcal{L}_{\text{LC}}(\delta) = \frac{w_i}{\beta}\tanh(\beta\delta_i) \quad\Longrightarrow\quad \big(\partial_i\mathcal{L}_{\text{LC}}(\delta)\big)^2 = \frac{w_i^2}{\beta^2}\tanh^2(\beta\delta_i).$$

Since $\phi(a) := \frac{\tanh^2 a}{\log\cosh a}$ is decreasing and $|\delta_i| \leq E$ on the sublevel,

$$\tanh^2(\beta\delta_i) = \phi(\beta|\delta_i|)\,\log\cosh(\beta\delta_i) \;\geq\; \phi(\beta E)\,\log\cosh(\beta\delta_i).$$

Therefore

$$\sum_i \big(\partial_i\mathcal{L}_{\text{LC}}(\delta)\big)^2 \;\geq\; \frac{\phi(\beta E)}{\beta^2}\sum_i w_i^2\log\cosh(\beta\delta_i) \;\geq\; \phi(\beta E)\,w_{\min}\,\mathcal{L}_{\text{LC}}(\delta),$$

where we used $w_i^2 \geq w_{\min}w_i$ and the definition of $\mathcal{L}_{\text{LC}}$. Finally, $(\sum_i |a_i|)^2 \geq \sum_i a_i^2$ gives $\|\nabla\mathcal{L}_{\text{LC}}(\delta)\|_1^2 \geq w_{\min}\,\phi(\beta R)\,\mathcal{L}_{\text{LC}}(\delta)$, and dividing by 2 yields the stated inequality.

*LSC.* Write $A(\delta) := \sum_j w_j \cosh(\beta \delta_j)$ and $p_i := \frac{w_i \cosh(\beta \delta_i)}{A}$. Then $\nabla \mathcal{L}_{\mathrm{LSC}}(\delta)_i = p_i \tanh(\beta \delta_i)$ and

$$\|\nabla \mathcal{L}_{\mathrm{LSC}}\|_1^2 \geq \sum_i p_i^2 \tanh^2(\beta \delta_i) \geq p_{\min} \sum_i p_i \tanh^2(\beta \delta_i).$$

As above, $\tanh^2(\beta \delta_i) = \phi(\beta |\delta_i|) \log \cosh(\beta \delta_i) \geq \phi(\beta R) \log \cosh(\beta \delta_i)$. Hence

$$\sum_i p_i \tanh^2(\beta \delta_i) \geq \phi(\beta R) \sum_i p_i \log \cosh(\beta \delta_i).$$

Using the log-sum-exp variational identity with $x_i := \log \cosh(\beta \delta_i)$,

$$\beta \, \mathcal{L}_{\mathrm{LSC}}(\delta) = \log \sum_i w_i e^{x_i} = \sum_i p_i x_i - \mathrm{KL}(p \| w) \leq \sum_i p_i x_i,$$

so $\sum_i p_i \log \cosh(\beta \delta_i) \geq \beta \, \mathcal{L}_{\mathrm{LSC}}(\delta)$. Putting these together: $\|\nabla \mathcal{L}_{\mathrm{LSC}}\|_1^2 \geq \beta \, p_{\min} \, \phi(\beta R) \, \mathcal{L}_{\mathrm{LSC}}$. Divide by 2 to match the stated form. □

*Remark* C.7 (On $p_{\min}$ and shaping via $\beta$). For LSC, $p(\delta)$ depends on $\beta$ through $p_i \propto w_i \cosh(\beta \delta_i)$. More generally, choosing $\beta$ to keep $\beta R$ small (early in training) keeps $p$ well-shaped relative to $w$; as $R$ decreases along optimization, $\beta$ can be increased while maintaining $\beta R = O(1)$.

**Step 5: Conclusion—contraction and step budgets for the $\alpha$-condition**    Combining the descent inequality (Lemma C.2) with the $\ell_1$–PL bounds (Lemma C.6) gives a linear contraction on the sublevel set $\{\|\delta\|_\infty \leq R\}$:

$$\mathcal{L}(\delta_{k+1}) \leq \left(1 - \frac{\mu_\infty}{L_\infty}\right) \mathcal{L}(\delta_k), \qquad \Longrightarrow \qquad \mathcal{L}(\delta_K) \leq \exp\!\left(-\frac{\mu_\infty}{L_\infty} K\right) \mathcal{L}(\delta_0).$$

Thus to reach an $\alpha$-reduction, $\mathcal{L}(\delta_K) \leq \alpha \, \mathcal{L}(\delta_0)$, it suffices to take

$$K \geq \frac{L_\infty}{\mu_\infty} \log \frac{1}{\alpha}.$$

Instantiating $L_\infty$ from Lemma C.1 and $\mu_\infty$ from Lemma C.6 yields

$$\frac{L_\infty}{\mu_\infty} = \begin{cases} \dfrac{1}{w_{\min}}, & \text{MSE}, \\[2mm] \dfrac{2}{w_{\min} \, \phi(\beta R)}, & \text{LC}, \\[2mm] \dfrac{4}{p_{\min}(\delta) \, \phi(\beta R)}, & \text{LSC}. \end{cases}$$

Using the surrogate-specific $\alpha$-conditions $\alpha_{\mathrm{MSE}} = w_{\min} \left(\frac{1-\gamma}{1+\gamma}\right)^2$, $\alpha_{\mathrm{LC}} = w_{\min} \left(\frac{1-\gamma}{1+\gamma}\right)$, $\alpha_{\mathrm{LSC}} = \left(\frac{1-\gamma}{1+\gamma}\right)$, we obtain the step budgets

$$\boxed{\begin{aligned} K_{\mathrm{MSE}} &\geq \frac{1}{w_{\min}} \log \frac{1}{\alpha_{\mathrm{MSE}}} = \frac{1}{w_{\min}} \left(\log \frac{1}{w_{\min}} + 2 \log \frac{1+\gamma}{1-\gamma}\right), \\[2mm] K_{\mathrm{LC}} &\geq \frac{2}{w_{\min} \, \phi(\beta R)} \log \frac{1}{\alpha_{\mathrm{LC}}} = \frac{2}{w_{\min} \, \phi(\beta R)} \left(\log \frac{1}{w_{\min}} + \log \frac{1+\gamma}{1-\gamma}\right), \\[2mm] K_{\mathrm{LSC}} &\geq \frac{4}{p_{\min}(\delta) \, \phi(\beta R)} \log \frac{1}{\alpha_{\mathrm{LSC}}} = \frac{4}{p_{\min}(\delta) \, \phi(\beta R)} \log \frac{1+\gamma}{1-\gamma}. \end{aligned}}$$

By coercivity (Lemma C.4) and monotone descent, the iterates remain in the bounded sublevel set of $S_0 := \mathcal{L}(\delta_0)$, so these constants are valid along the entire inner-loop trajectory. In practice one can shape $p$ via $\beta$ so that with a "moderate" radius (e.g. $\beta R \approx 1$) and maintaining $\beta R = O(1)$ keeps $p$ well-shaped relative to $w$.

## D. Q-Network Architectures: MLP (MuJoCo) and CNN (Atari)

**Notes.**    The architectures are directly adopted from the CleanRL DQN implementation (Huang et al., 2022); in Atari CNN the forward pass divides inputs by 255.0 (see code in the main text). The MuJoCo policy uses a 3-layer MLP on flattened observations with ReLU nonlinearities.

**Compute.**    All experiments ran on a single NVIDIA GPU on a standard Linux server (e.g., A100, V100, RTX8000); no multi-GPU, TPU, or other specialized hardware was used.

*Table 1.* Observation preprocessing and input formatting used in our experiments. The Atari setup follows the common DQN-style pipeline; the MuJoCo setup uses raw vector observations flattened to a single dimension.

| Hyper-parameter | Atari (pixels) | MuJoCo (state vectors) |
|---|---|---|
| Observation type | Stacked frames (4) | Low-dimensional state |
| Input shape to network | (4, 84, 84) | $(d)$ (flattened) |
| Scaling / normalization | Divide by 255.0 | None (raw values) |
| Frame stacking | 4 | N/A |
| Color space | Grayscale | N/A |
| Flatten before FC | After conv stack | Input already flat |

*Table 2.* Q-network architectures. Let $|A|$ denote the number of discrete actions and $d = \prod_i \mathrm{shape}(s)_i$ the flattened MuJoCo observation dimension. All hidden layers use ReLU activations; the output heads are linear (no activation).

| Domain | Architecture (layer $\rightarrow$ units / kernel, stride) |
|---|---|
| Atari (CNN) | Conv2d $(4 \rightarrow 32, 8 \times 8, \mathrm{stride} = 4)$, ReLU |
| | Conv2d $(32 \rightarrow 64, 4 \times 4, \mathrm{stride} = 2)$, ReLU |
| | Conv2d $(64 \rightarrow 64, 3 \times 3, \mathrm{stride} = 1)$, ReLU |
| | Flatten |
| | Linear $(3136 \rightarrow 512)$, ReLU |
| | Linear $(512 \rightarrow |A|)$ |
| MuJoCo (MLP) | Linear $(d \rightarrow 120)$, ReLU |
| | Linear $(120 \rightarrow 84)$, ReLU |
| | Linear $(84 \rightarrow |A|)$ |
| | (expects input of shape $(B, d)$; outputs $(B, |A|)$) |

## E. Additional Experiments

### E.1. Ablation on Model Size

We study how function-approximation capacity interacts with surrogate choice following the same setup as Schwarzer et al. (2023) by scaling the standard Nature-DQN convolutional architecture via a width multiplier $\tau \in \{0.5, 1, 2\}$. Model architecture is presented in Table 3. Concretely, we multiply the convolutional channel counts $(32, 64, 64)$ and the fully-connected hidden width 512 by $\tau$ (rounded to integers, with a minimum of 1 channel/unit), yielding networks whose parameter count scales approximately as $\Theta(\tau^2)$. All other hyperparameters are kept fixed, and we evaluate on DemonAttack, Phoenix, and MsPacman using greedy evaluation returns.

Figure 6, 7, 8 shows that increased capacity typically improves performance across games, with the clearest gains appearing once $K$ is moderate/large (50/500). This is consistent with the surrogate perspective: once the inner loop can make meaningful progress against a frozen target, performance becomes increasingly limited by approximation error rather than optimization. We also observe some non-monotonicity with increasing $\tau$ in a few configurations (notably DemonAttack), which is consistent with the fact that off-policy bootstrapping can amplify optimization noise and replay-induced distribution shift, so higher-capacity Q network do not always translate into higher returns under fixed hyperparameters.

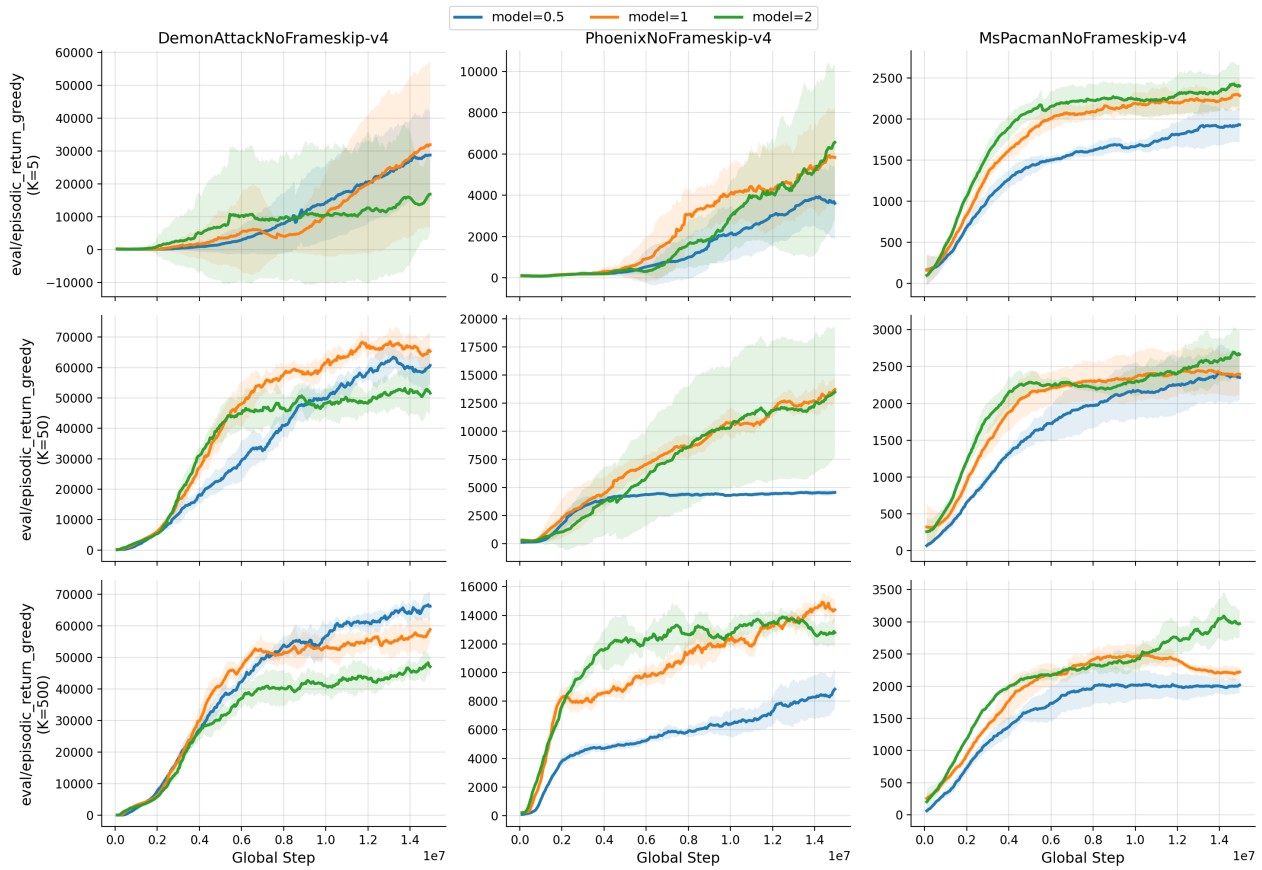

*Figure 6.* Episodic return of MSE with same number of inner steps and different model size. Shaded bands are std over seeds.

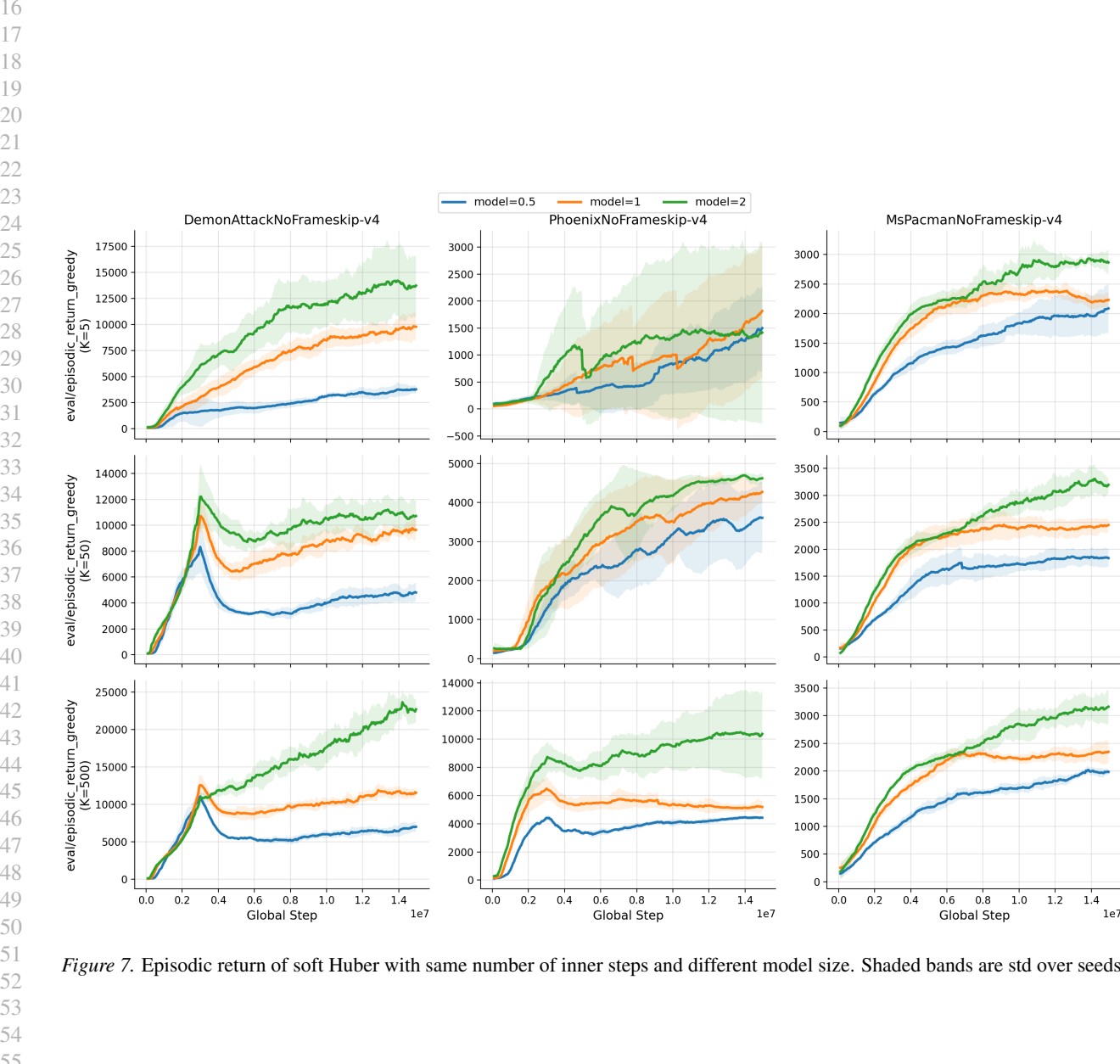

*Figure 7.* Episodic return of soft Huber with same number of inner steps and different model size. Shaded bands are std over seeds.

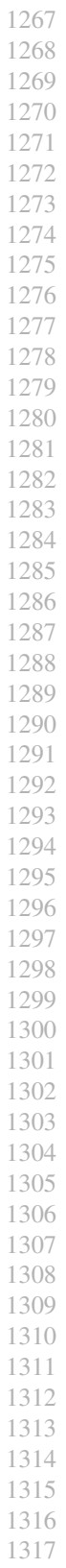
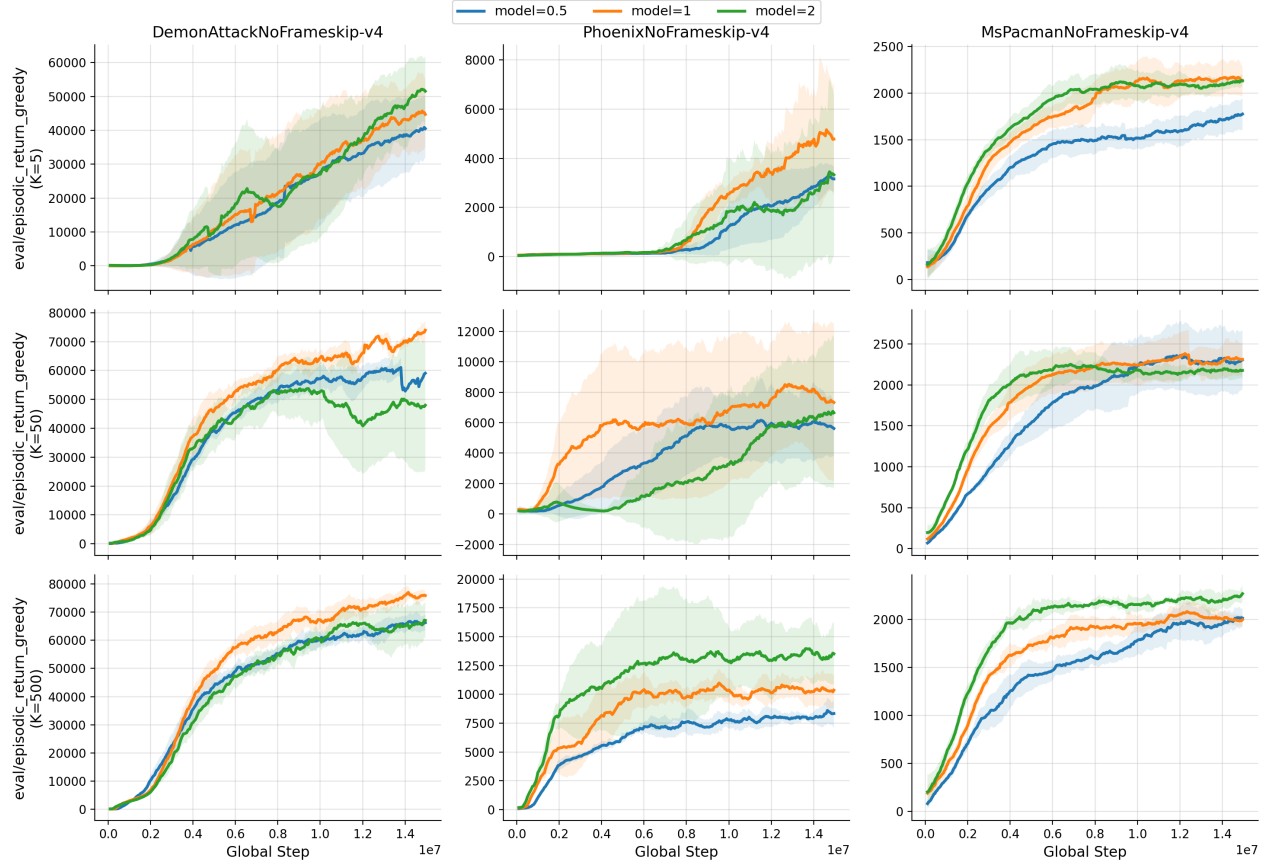

*Figure 8.* Episodic return of soft $\ell_\infty$ with same number of inner steps and different model size. Shaded bands are std over seeds.

*Table 3.* **Atari Q-network (Nature DQN) with width scaling.** We scale model capacity by a width multiplier $\tau \in \{0.5, 1, 2\}$. Channel and hidden widths are rounded to the nearest integer and clipped to be at least 1.

| Component | Specification (layer $\rightarrow$ channels/units, kernel, stride) |
|---|---|
| Input | $84 \times 84$ grayscale, 4 stacked frames |
| Conv1 | Conv2d $(4 \rightarrow \lfloor 32\tau \rceil, 8 \times 8, \text{stride} = 4)$, ReLU |
| Conv2 | Conv2d $(\lfloor 32\tau \rceil \rightarrow \lfloor 64\tau \rceil, 4 \times 4, \text{stride} = 2)$, ReLU |
| Conv3 | Conv2d $(\lfloor 64\tau \rceil \rightarrow \lfloor 64\tau \rceil, 3 \times 3, \text{stride} = 1)$, ReLU |
| Flatten | $7 \times 7$ spatial map $\Rightarrow$ features $= 7 \cdot 7 \cdot \lfloor 64\tau \rceil$ |
| FC | Linear $(7 \cdot 7 \cdot \lfloor 64\tau \rceil \rightarrow \lfloor 512\tau \rceil)$, ReLU |
| Output | Linear $(\lfloor 512\tau \rceil \rightarrow |A|)$ |

## E.2. Ablation on Inner Steps

We next ablate the number of optimization steps performed per frozen target network, which corresponds to the inner-loop budget $K_{\text{target}}$ in Algorithm 1 on the same three Atari environments.

Figure 9, 10, 11 demonstrates that increasing $K_{\text{target}}$ is a primary driver of performance on DemonAttack and Phoenix, while MsPacman exhibits earlier saturation. This aligns with the role of the $\alpha$-condition in Theorem 3.3: larger $K_{\text{target}}$ gives the inner loop sufficient budget to reduce the surrogate loss against a fixed target before the next target refresh, thereby better realizing the predicted outer-loop improvement behavior. Across surrogates, the qualitative dependence on $K_{\text{target}}$ is consistent: small $K_{\text{target}}$ can lead to under-optimized regression against rapidly changing targets, whereas moderate/large $K_{\text{target}}$ substantially stabilizes learning and improves returns.

## E.3. Ablation on Fixed Temperature

**Findings.** Across all games, **soft-$\ell_\infty$ with adaptive** $\beta$ is consistently competitive with, or superior to, any fixed choice of $\beta$, supporting the use of a controller that increases worst-case focus as training stabilizes. For **soft-Huber**, smaller $\beta$ (e.g., $\beta = 1$) is mostly favored, which matches the MSE behavior: a smaller $\beta$ enlarges the quadratic (Euclidean) region of the loss, yielding updates closer to $\ell_2$ regression early on. These observations suggest that *adaptive $\beta$ is particularly impactful for soft-$\ell_\infty$*, where alignment with the $\ell_\infty$ control geometry improves as $\beta$ grows, while the necessity of adapting $\beta$ for soft-Huber remains unclear in our Atari ablations, though concurrent work proposes dynamic Huber schedules as beneficial (Xu et al., 2025).

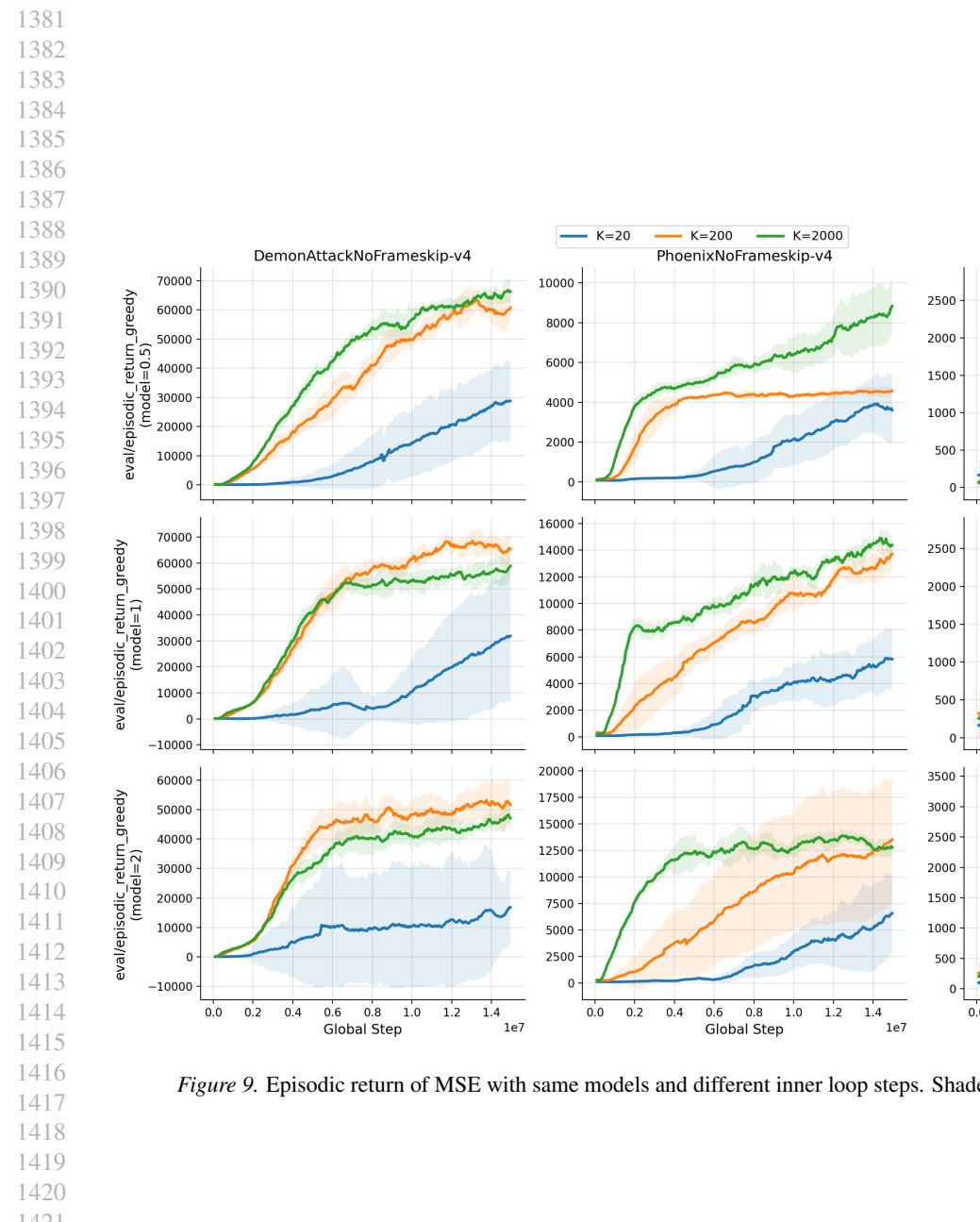

*Figure 9.* Episodic return of MSE with same models and different inner loop steps. Shaded bands are std over seeds.

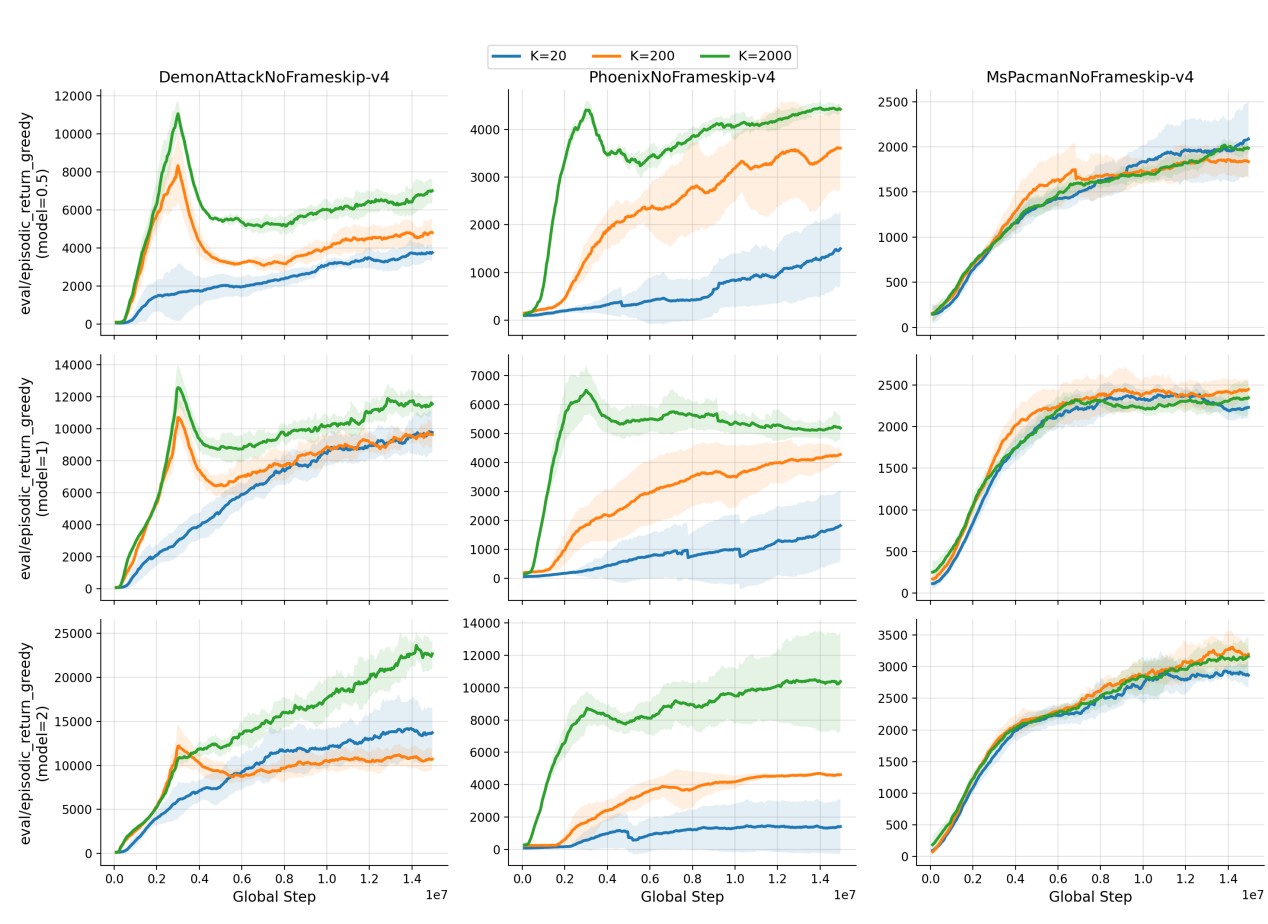

*Figure 10.* Episodic return of soft Huber with same models and different inner loop steps. Shaded bands are std over seeds.

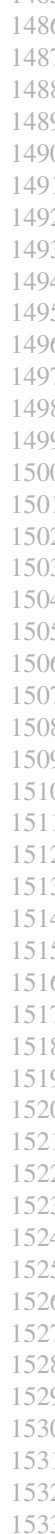
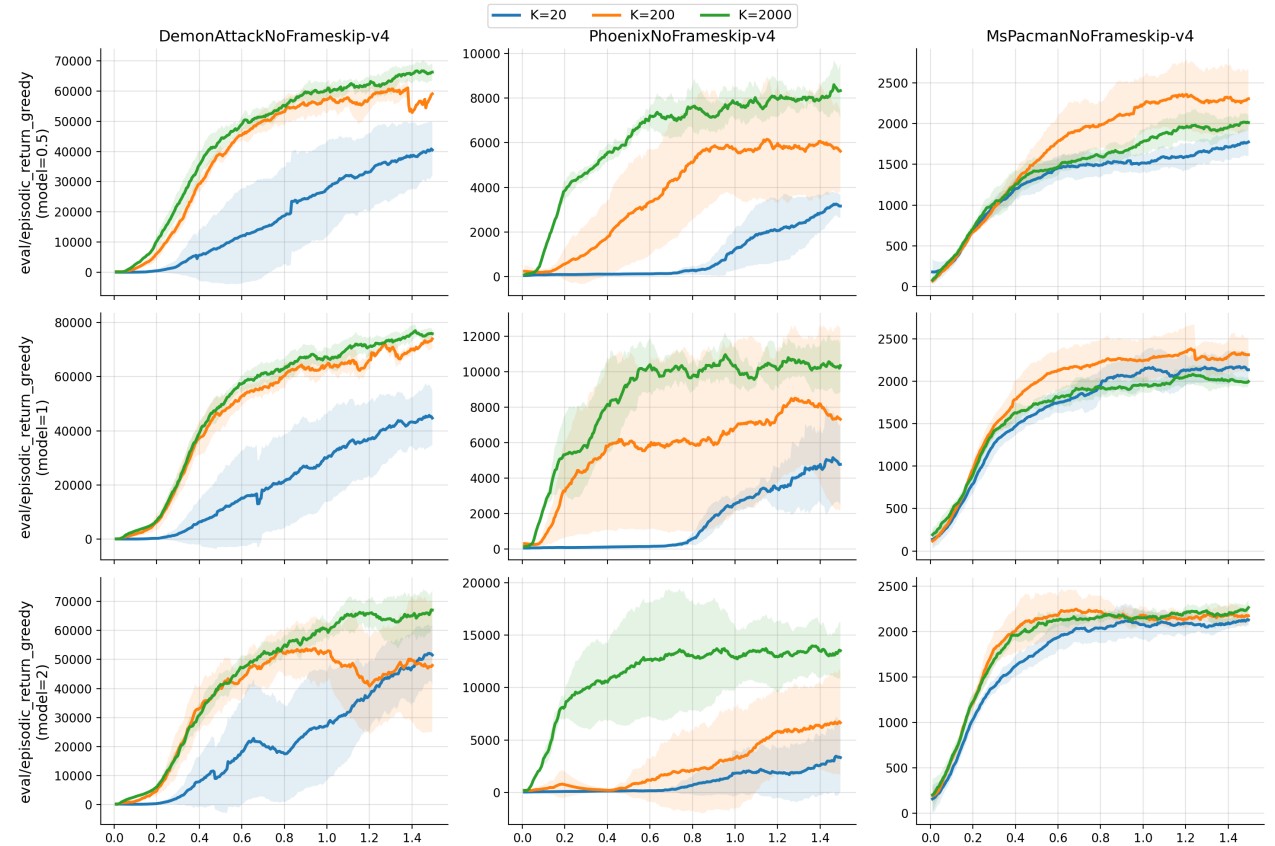

*Figure 11.* Episodic return of soft $\ell_\infty$ with same models and different inner loop steps. Shaded bands are std over seeds.

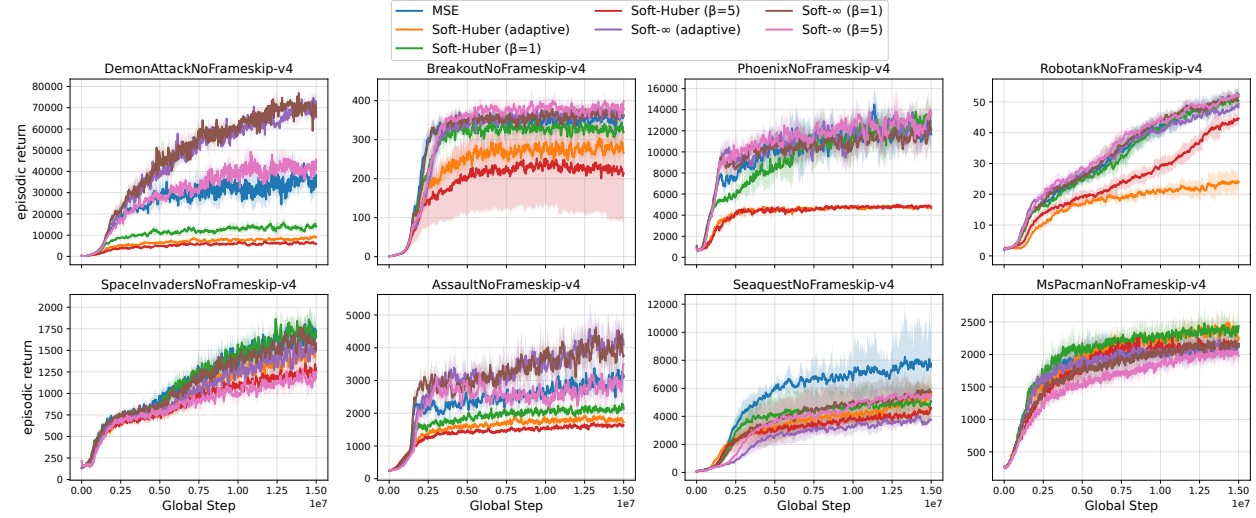

*Figure 12.* **Fixed-$\beta$ ablation vs. adaptive $\beta$.** Episodic return on eight Atari games comparing soft-$\ell_\infty$ and soft-Huber with *fixed* temperatures ($\beta \in \{1, 5\}$) against their *adaptive* variants; MSE shown for reference. Shaded bands are std over seeds.

