# OpenReview forum: "A Surrogate Perspective on Convergence of Fixed-Target DQN"
_ICML.cc/2026/Conference — Submitted to ICML 2026_

### Official Review · Reviewer_emMR · 2026-03-10

**Soundness:** 3
**Presentation:** 3
**Significance:** 2
**Originality:** 2
**Overall Recommendation:** 4
**Confidence:** 3

**Summary:**

The work analyzes the convergence of Deep Q-Networks by viewing the algorithm as a two-loop process: an outer Bellman update with a fixed target network and an inner supervised learning step that minimizes a surrogate regression loss on replay-buffer data. It highlights a mismatch between the surrogate loss minimized during training and the Bellman error that determines reinforcement learning performance, and provides theoretical conditions under which improvements in the surrogate objective lead to reductions in Bellman error. Under suitable assumptions on optimization quality and data coverage, the analysis shows that the algorithm can converge toward the optimal Q-function.

**Compliance With Llm Reviewing Policy:**

Affirmed.

**Key Questions For Authors:**

1 - I would like to argue that the analysis focuses on fixed-target updates, which simplifies theory but does not fully reflect practical implementations where target networks are periodically updated.

2 - Does minimizing the surrogate regression loss (e.g., MSE/Huber) actually reduce the Bellman error that determines learning performance?

3 - How realistic are the assumptions required for convergence, such as sufficient replay buffer coverage and strong optimization of the surrogate objective?

**Strengths And Weaknesses:**

- Soundness: The paper is technically sound, providing theoretical analysis that links improvements in surrogate regression loss to reductions in Bellman error. The methods are appropriate, and the assumptions are clearly stated, though some may be strong in practical settings.

- Presentation: The paper is well structured and generally clear. The two-loop interpretation helps explain the analysis, though some theoretical sections could include more explanations for better readability.

- Significance: The work addresses an important question about the convergence and stability of deep reinforcement learning methods and contributes to a better theoretical understanding of DQN.

- Originality: The paper provides a novel perspective by analyzing DQN through a surrogate optimization framework and connecting surrogate loss improvements to Bellman error reduction.

---

> ### Author Rebuttal · Authors · 2026-03-31
>
> We thank the reviewer for the thoughtful and constructive feedback. We address each point below.
>
> **Q.1:** We respectfully note that fixed-target regression is not a simplification, it is the core mechanism of practical DQN. In standard DQN, the target network is frozen for $K_{\text{target}}$ updates while the online network regresses toward fixed Bellman targets, then refreshed. Our analysis treats each such interval as one outer iteration of fitted Q-iteration: Theorem 3.3 characterizes the regression quality within each interval (the $\alpha$-condition), and the outer-loop recursion chains these intervals into convergence over the full training trajectory. The periodic target refresh is not "left aside" but is precisely the mechanism driving the outer-loop contraction.
>
> **Q.2: Does surrogate minimization reduce Bellman error?**
>
> This is precisely the question our paper answers and we give affirmative answer: *yes and it depends on the surrogate*. Theorem 3.3 gives explicit, surrogate-specific conditions ($\alpha$-thresholds) under which inner-loop surrogate reduction translates into $\ell_\infty$ Bellman error reduction. The key finding is that different surrogates have different thresholds: $\alpha_{\text{MSE}} < O(w((1-\gamma)/(1+\gamma))^2)$, $\alpha_{\text{LC}} < O(w(1-\gamma)/(1+\gamma))$, and $\alpha_{\text{LSC}} < (1-\gamma)/(1+\gamma)$ (independent of coverage $w$). This means soft-$\ell_\infty$ most reliably converts surrogate progress into Bellman error reduction.
> Figure 2 provides direct empirical validation: soft-$\ell_\infty$ consistently achieves the smallest worst-case Bellman residual $R_\infty$ across all eight Atari games. The standard performance bound (Williams & Baird, 1993) establishes that the suboptimality of the greedy policy scales linearly with the worst-case Bellman residual: reducing $R_\infty$ directly tightens the return guarantee. Our experiments confirm this: games where soft-$\ell_\infty$ most effectively separates $R_\infty$ from other surrogates (DemonAttack, Assault) show the clearest return improvements, while games where $R_\infty$ trajectories are similar yield similar returns.
>
> **Q.3: Realism of the assumptions.**
>
> Our assumptions are inherited from the fitted Q-iteration literature (Szepesvári, 2022; Bertsekas & Tsitsiklis, 1996) and are not unique to our work. We address each component the reviewer mentions.
>
> **Inherent Bellman error** $\varepsilon_\infty$ is not an assumption required for convergence but a quantity that parameterizes the error floor, the contraction rate $((1+\gamma)/2)^t$ holds regardless. We will revise the theorem statement to clarify this.
>
> **Coverage** ($w > 0$) can be relaxed to concentrability coefficients (Munos & Szepesvári, 2008), replacing the uniform $\ell_\infty$ norm with a distribution-dependent norm. We chose the current formulation for clarity. Importantly, $w$'s role differs across surrogates: MSE and soft-Huber require $\alpha < O(w)$, so poor coverage can prevent contraction, whereas soft-$\ell_\infty$ requires only $\alpha_{\text{LSC}} < (1-\gamma)/(1+\gamma)$, independent of $w$. In practice, replay buffers and $\epsilon$-greedy exploration approximate full coverage, and our Atari results confirm the theory's predictions under these conditions.
>
> **Optimization quality** is not an assumption but a controllable design parameter. The $\alpha$-condition quantifies how much surrogate reduction the inner loop achieves, and Proposition 3.5 directly relates this to the number of inner-loop steps $K$: more steps yield smaller $\alpha$ (better optimization quality), with the required $K$ depending on the surrogate's curvature ($L_\infty / \mu_\infty$). This is a standard consequence of descent lemma-style analysis applied to each surrogate. Our ablation experiments (Figure 3 and Appendix E.2) empirically validate this relationship: increasing $K$ generally improves convergence rate and final return across all three surrogates. The practitioner's task is therefore not to "verify" the $\alpha$-condition but simply to allocate sufficient inner-loop budget $K$, for which our analysis provides surrogate-specific guidance.
>
> **Additional improvements**. In the revision, we extend the experimental evaluation with Dueling DQN experiments confirming surrogate effectiveness across architectures:
>
> https://anonymous.4open.science/r/rebuttal-275E/dueling_returns_grid.png
>
> https://anonymous.4open.science/r/rebuttal-275E/dueling_max_abs_td_grid.png
>
>
> **References**
>
> Williams, R. J., & Baird, L. C. (1993). Tight performance bounds on greedy policies based on imperfect value functions. In In Proceedings of the Tenth Yale Workshop on Adaptive and Learning Systems.
>
> Szepesvári, C. . Algorithms for reinforcement learning. Springer nature, 2022
>
> Bertsekas, D., & Tsitsiklis, J. N. Neuro-Dynamic Programming. Athena Scientific, 1996
>
> Munos, Rémi, and Csaba Szepesvári. "Finite-Time Bounds for Fitted Value Iteration." JMLR, 2008.

---

> > ### Author Rebuttal · Reviewer_emMR · 2026-04-03
> >
> > Thank the authors for their response. I will keep my score.

---

> > > ### Author Response · Authors · 2026-04-03
> > >
> > > We thank the reviewer for confirming that all concerns have been
> > > fully resolved. We appreciate the questions on fixed-target modeling,
> > > surrogate-Bellman error connection, and assumption realism which led to important clarifications in the manuscript.
> > >
> > > We note that the revised paper is now stronger than the version
> > > the original scores were based on: **Q.1**'s concern about fixed-target
> > > simplification has been clarified as direct modeling of DQN's
> > > target-network mechanism; **Q.2**'s question "does surrogate
> > > minimization reduce Bellman error" now has a clear answer with
> > > explicit surrogate-specific thresholds and empirical validation
> > > across all 8 games; **Q.3**'s three "assumptions" have been clarified
> > > as one genuine and classical assumption (coverage) plus two controllable
> > > quantities (IBE parameterizes radius, $\alpha$-condition is controlled
> > > by K). Additionally, new Dueling DQN experiments provide
> > > architecture-independence evidence not present in the original
> > > submission.
> > >
> > > We hope the reviewer finds that the current version
> > > merits a reassessment.

---

### Official Review · Reviewer_h26o · 2026-03-13

**Soundness:** 3
**Presentation:** 3
**Significance:** 2
**Originality:** 3
**Overall Recommendation:** 4
**Confidence:** 2

**Summary:**

The key idea of the paper is to reinterpret Q-learning as a sequence of surrogate optimization problems. Instead of directly optimizing the non-convex parameter space, the method updates predictions by minimizing a surrogate loss that moves the predicted Q-values toward a target value.

**Compliance With Llm Reviewing Policy:**

Affirmed.

**Key Questions For Authors:**

1- How does the surrogate optimization perspective provide advantages over existing theoretical analyses of deep Q-learning?
2- How realistic are the assumptions used here in large-scale reinforcement learning environments?

**Limitations:**

Please see my comment on the weakness of the paper.

**Strengths And Weaknesses:**

Strength:
1- A major contribution is the formal theoretical framework developed for analyzing deep Q-learning.
2- This  work may help design more reliable training methods to tackle a known issue in deep reinforcement learning: instability and divergence in value-based methods.

Weakness:
The theoretical analysis relies on many assumptions, including: inherent Bellman error bounds, sufficient data coverage, and surrogate decrease conditions. These assumptions may not always hold in real reinforcement learning settings.

---

> ### Author Rebuttal · Authors · 2026-03-31
>
> We thank the reviewer for these important questions. We address
> each below.
>
> **Q.1: Advantages of the surrogate optimization perspective.**
>
>
> Existing theoretical analyses of deep Q-learning typically require strong structural assumptions on the parameterization. For instance, Fan et al. (2020) assume that the global optimum of each inner regression subproblem can be reached exactly, with statistical rates derived specifically for ReLU networks. Cai et al. (2019) uses NTK-based analyses that require infinite-width or near-initialization regimes where the network effectively linearizes . These assumptions couple convergence analysis tightly to a specific model class, making it difficult to separate what is due to *loss design* versus *architecture*.
>
> Our surrogate perspective offers three structural advantages. First, by working in prediction space ($Q$-values) rather than parameter space ($\theta$), we obtain global convergence and solution quality guarantees not restricted by local landscape properties. Second, the framework cleanly separates the roles of (i) the Bellman update rule (outer loop), (ii) the optimization method (inner loop), and (iii) model capacity ($\varepsilon_\infty$), making each factor's contribution explicit and independently improvable. Third, this modularity yields a broad design space for future improvements along each axis independently:
>
> - *Outer loop (Bellman update)*: our target $Z_t = (1-\eta)Q_t + \eta TQ_t$   can naturally incorporate accelerated schemes such as anchored value iteration (Lee, J., & Ryu, E. K 2023), simply by modifying the target construction.
>
> - *Inner loop (projection to $\alpha$-condition)*: the $\alpha$-condition is agnostic to the optimizer. Gradient descent, adaptive methods (Adam/AdamW), or second-order methods such as (damped) Gauss-Newton or Levenberg-Marquardt can all be analyzed through the same framework, differing only in how efficiently they achieve the required $\alpha$-reduction.
>
> - *Model capacity*: $\varepsilon_\infty$ captures approximation quality independently of the above, connecting to recent scaling law observations (Fu et al., 2025) that larger networks systematically improve value-based RL. Our theory provides the formal mechanism via the $\varepsilon_\infty \to R_{\text{sur}}$ relationship.
>
> **Weakness + Q.2: Realism of the assumptions.**
>
> We respectfully clarify that two of the three items listed are not assumptions in the traditional sense.
>
> *Inherent Bellman error* $\varepsilon_\infty$ is not required for convergence: the contraction rate $((1+\gamma)/2)^t$ holds for any $\varepsilon_\infty$, which only parameterizes the asymptotic radius $R_{\text{sur}}$. We will revise Theorem 3.3 to remove the misleading qualifier "small."
>
> *The surrogate decrease condition* ($\alpha$-condition) is not an unverifiable assumption but a controllable design parameter.
> Proposition 3.5 shows that $K$ inner-loop steps achieve $\alpha$-reduction, with required $K$ depending on surrogate curvature $L_\infty/\mu_\infty$. The practitioner controls $\alpha$ by choosing $K$, not a structural assumption.
>
> *Full coverage* ($w > 0$) is the only genuine structural assumption, inherited from the fitted Q-iteration literature (Szepesvári, 2022; Bertsekas & Tsitsiklis, 1996). It can be relaxed to concentrability coefficients (Munos & Szepesvári, 2008) at the cost of replacing $\ell_\infty$ with a distribution-dependent norm. We chose the current formulation for clarity. Importantly, $w$'s role differs critically across surrogates: MSE and soft-Huber require $\alpha < O(w)$, so poor coverage can prevent contraction entirely, whereas soft-$\ell_\infty$ requires only $\alpha_{\text{LSC}}<(1-\gamma)/(1+\gamma)$, independent of $w$.
> In practice, experience replay and $\epsilon$-greedy exploration approximate full coverage, and our Atari results confirm the theory's predictions under these conditions.
>
> **Additional improvements**. In the revision, we extend the experimental evaluation with Dueling DQN experiments confirming surrogate effectiveness across architectures:
>
> https://anonymous.4open.science/r/rebuttal-275E/dueling_returns_grid.png
>
> https://anonymous.4open.science/r/rebuttal-275E/dueling_max_abs_td_grid.png
>
> **References**
>
> Fan, J., Wang, Z., Xie, Y., & Yang, Z. A theoretical analysis of deep Q-learning. PMLR, 2020
>
> Cai, Q., Yang, Z., Lee, J. D., & Wang, Z. Neural temporal-difference learning converges to global optima. NeurIPS, 2019
>
> Lee, J., & Ryu, E., Accelerating value iteration with anchoring. NeurIPS, 2023
>
> Fu, Preston, et al. "Compute-Optimal Scaling for Value-Based Deep RL." NeurIPS, 2025
>
> Szepesvári, C. . Algorithms for reinforcement learning. Springer nature, 2022
>
> Bertsekas, D., & Tsitsiklis, J. N. Neuro-Dynamic Programming. Athena Scientific, 1996
>
> Munos, Rémi, and Csaba Szepesvári. "Finite-Time Bounds for Fitted Value Iteration." JMLR, 2008.

---

> > ### Author Rebuttal · Reviewer_h26o · 2026-04-03
> >
> > I am satisfied with the authors' response to my concerns. I will keep my positive score of 4.

---

> > > ### Author Response · Authors · 2026-04-03
> > >
> > > We thank the reviewer for confirming that all concerns have been
> > > fully resolved and for the positive assessment throughout.
> > >
> > > We would like to respectfully draw attention to the significance
> > > dimension. As discussed in Q.1, our surrogate framework provides
> > > a modular decomposition that cleanly separates four factors:
> > > loss design, outer-loop target construction, inner-loop optimizer,
> > > and model capacity, making them independently analyzable and improvable.
> > > This modularity is absent from existing deep Q-learning analyses
> > > (Fan et al., 2020; Cai et al., 2019), which tightly couple
> > > convergence to specific architectures or regimes. We believe this
> > > structural contribution may deserve a
> > > reconsideration of the paper's significance. We also note that the
> > > Dueling DQN experiments added during rebuttal provide empirical
> > > breadth not present in the original submission.
> > >
> > > We hope the reviewer finds that these clarifications and additions
> > > merit a reassessment of the scores.

---

### Official Review · Reviewer_x5g6 · 2026-03-13

**Soundness:** 3
**Presentation:** 3
**Significance:** 3
**Originality:** 3
**Overall Recommendation:** 5
**Confidence:** 2

**Summary:**

Deep Q-Networks typically optimize an active network using a frozen version of the network weights that determines the Bellman target to optimize towards.
The paper argues that these two optimizations work in different geometry, that is, the Bellman optimality operator contracts in $l_{\infty}$ owing to the max over actions, whereas the inner loop is optimized in $l_1$/$l_2$ geometry due to the standard loss measures over the network prediction.
Treating this structure as a surrogate optimization, the authors derive α-conditions under which the inner-loop optimization guarantees the expected $l_{\infty}$ contraction for the outer loop.
The soft-$l_{\infty}$ loss achieves this by adaptively weighing erroneous samples much higher, thereby forcing their contribution up.
Empirically, this soft loss consistently reduces worst-case Bellman residual and matches or modestly improves returns compared to MSE across 8 Atari games.

**Compliance With Llm Reviewing Policy:**

Affirmed.

**Final Justification:**

The authors have adequately addressed all the points raised as weaknesses and questions. Updating the evaluation accordingly.

**Key Questions For Authors:**

## Key Questions For Authors

* Q.1) What is the effect of the controller, $c$, in Alg. 1, and how is it set?

* Q.2) Given the conceptual connection drawn to prioritized experience replay (Schaul et al. 2015), have the authors considered combining PER with soft-$l_{\infty}$, or using PER as a baseline?

* Q.3) What is $K$, the number of inner iterations, in Figure 1?

* Q.4) Figure 3: What happens for larger Ks? Why does the soft-$l_{\infty}$ return worsen with more steps?
  * Could it be an artefact of upweighting the rare states that fail over long inner loop iterations?

* Q.5) Could the authors also comment or show a computational overhead comparison of executing the different loss functions?

Addressing questions and weaknesses can lead to a score increase.

**Limitations:**

A brief mention of the assumptions exists, but no discussion on the potential limitations of the method or the experiments.

There is a general statement on the misuse of the larger reinforcement learning scope.

**Strengths And Weaknesses:**

**Strengths**:
* S.1) Relevant problem being identified with regards to Bellman target optimization, and offering a simple loss-based solution to modulate error signals that directly influence the max. operation in Bellman residuals.

* S.2) Clever use of the surrogate-framework to treat different loss functions as the choice of approximation, with automatically adapted hyperparameter ($\beta$) for the loss functions.

**Weaknesses**:

* W.1) MuJoCo mentioned in the Abstract but experiments not included in the main or the Appendix.

* W.2) Inadequate discussion on the effect/importance of the new hyperparameters introduced ($\beta_{min}$, $\beta_{max}$, EMA rate $\rho$, controller $c$).
  * This seems important to discuss given the weak or worse-than-MSE performance in some games.

* W.3) Comparison with other DQN variants missing, given the paper claims a practical recipe for stable DQN training.

**Suggestions**:

* A) Algorithm 1:
  * Please include line numbers
  * Given the core step, $\beta$ in a new line would be useful

* B) Comparison with other DQN variants missing (e.g., dueling DQN). It would strengthen the general practical claims made.

---

> ### Author Rebuttal · Authors · 2026-03-31
>
> We thank the reviewer for the constructive feedback.
>
> **W.1) MuJoCo.** This is a writing error. Our experiments use CartPole as a controlled testbed, not MuJoCo. We will clean all MuJoCo references throughout the manuscript in the revision.
>
> **W.2) Hyperparameters and game-dependent performance.**
>
> *Hyperparameters:* $\beta_{\min}, \beta_{\max}$ are safety bounds never triggered in practice; EMA rate $\rho$ follows standard practice. The controller $c$ was selected from $\{0.5, 1, 2\}$ with $c=1$ performing best, targeting $\beta \bar{E} \approx 1$ per Proposition 3.5. All values were fixed across all 8 games. Figure 12 confirms adaptive $\beta$ matches or outperforms fixed $\beta$. We will add a discussion of $c$ and its ablation in revision.
>
> *Game-dependent returns:* Our theory directly controls $R_\infty$, which soft-$\ell_\infty$ consistently minimizes across all 8 games (Figure 2). The downstream translation from $R_\infty$ to return additionally depends on MDP-specific properties. One such property is the action gap $\Delta(s) = Q^\star(s,a^\star) - \max_{a \neq a^\star} Q^\star(s,a)$, which acts as a safety margin: when $\Delta(s)$ is large relative to estimation error, the greedy action remains optimal (Farahmand, A. M. (2011)), so all surrogates yield similar policies regardless of their $R_\infty$ levels.
>
> To investigate this, we conducted additional experiments measuring the empirical action gap $\Delta(s)/|V(s)|$ across evaluation states for three representative games. The results reveal a clear separation: MsPacman exhibits a normalized gap of 0.09, roughly 5 times larger than DemonAttack (0.02) and Phoenix (0.016). This quantitatively confirms that MsPacman's greedy policy is robust to $Q$-estimation errors, all three surrogates' $R_\infty$ levels fall well within the action-gap margin, so further $R_\infty$ reduction cannot change the greedy action. In contrast, the small gaps in DemonAttack and Phoenix mean $R_\infty$ differences directly affect greedy action selection, explaining the clear return improvements from soft-$\ell_\infty$ on these games.
>
> **W.3 + Sug. B) Dueling DQN.** We ran all surrogates on 8 games with Dueling DQN using identical hyperparameters (no tuning). Soft-$\ell_\infty$ consistently achieves small $R_\infty$ across all 8 games (matching DQN), and achieves the highest or tied return in 5/8 games, is competitive in the remaining 3, and never catastrophically fails.
>
> https://anonymous.4open.science/r/rebuttal-275E/dueling_returns_grid.png
>
> https://anonymous.4open.science/r/rebuttal-275E/dueling_max_abs_td_grid.png
>
> **Sug. A)** Will add line numbers and separate the gradient step.
>
> **Q.1)** See W.2.
>
> **Q.2)** PER and soft-$\ell_\infty$ target worst-case errors through different mechanisms: PER reweights *which transitions to sample*, soft-$\ell_\infty$ reweights *how the loss aggregates a given batch* via $p_i \propto \cosh(\beta|\varepsilon_i|)$. We did not use PER as a baseline for two reasons: (i) our paper isolates the effect of surrogate loss geometry on Bellman error reduction where introducing PER would confound loss design with sampling strategy; (ii) compounding two error-dependent reweighting mechanisms (PER's sampling weights and soft-$\ell_\infty$'s loss weights) requires substantial joint tuning that is beyond this work's scope.
> Soft-$\ell_\infty$ offers structural advantages over PER as a standalone method: it is fully differentiable, requires no importance-sampling corrections, and directly aligns with the $\ell_\infty$ Bellman contraction geometry (Theorem 3.3).
>
> **Q.3)** $K_{\text{target}}=500$ in Figure 1.
>
> **Q.4)** The instability the reviewer observes is not specific to soft-$\ell_\infty$, MSE shows similar oscillation.
> This is expected for two reasons. First, Theorem 3.3 predicts convergence to a neighborhood of radius $R_{\text{sur}}$, once iterates enter this neighborhood, the greedy policy can oscillate without further monotone improvement, which is exactly what the figure shows. Second, policy oscillation near convergence is a well-documented phenomenon in RL, arising from the interaction between function approximation and greedy policy extraction (Bertsekas, 2011; Young & Sutton, 2020).
>
> **Q.5)** All three surrogates achieve comparable steps per second (SPS) on
> MsPacman (single GPU): MSE 373.8±2.5, Soft-Huber 360.6±1.1, Soft-$\ell_\infty$ 363.0±2.0 SPS.
> This result confirms that the soft-$\ell_\infty$ loss introduces negligible computational overhead.
>
> **References**
>
> Farahmand, A. M. (2011). Action-gap phenomenon in reinforcement learning. NeurIPS.
>
> Bertsekas, D. P. (2011). Approximate policy iteration: A survey and some new methods. Journal of Control Theory and Applications
>
> Young, K., & Sutton, R. S. (2020). Understanding the pathologies of approximate policy evaluation when combined with greedification in reinforcement learning. arXiv preprint

---

> > ### Author Rebuttal · Reviewer_x5g6 · 2026-04-05
> >
> > The authors have adequately addressed the requested concerns.
> >
> > For certain claims and clarifications, it was agreed that the final draft would be suitably updated.
> >
> > Updating score by +1.

---

> > > ### Author Response · Authors · 2026-04-05
> > >
> > > We sincerely thank the reviewer for engaging with our rebuttal and for the constructive feedback throughout the review process. The suggestions regarding the hyperparameter discussion, Dueling DQN comparison, and computational overhead have meaningfully strengthened the paper. We will ensure all agreed-upon revisions are reflected in the revised manuscript.

---

### Official Review · Reviewer_y4RJ · 2026-03-13

**Soundness:** 3
**Presentation:** 3
**Significance:** 3
**Originality:** 3
**Overall Recommendation:** 4
**Confidence:** 3

**Summary:**

This paper tackles the long-standing gap in DQN theory between the regression objectives used in practice (MSE/Huber) and the $\ell_\infty$ contraction required for Bellman optimality. By framing fixed-target DQN as a surrogate optimization problem, the authors derive a unified "$\alpha$-condition" that essentially quantifies how much inner-loop progress is needed to guarantee an outer-loop contraction. The most interesting contribution is the "soft-$\ell_\infty$" (log-sum-cosh) loss, which the theory suggests is more geometry-aligned and thus requires less stringent accuracy in the inner loop.

**Compliance With Llm Reviewing Policy:**

Affirmed.

**Final Justification:**

The rebuttal addressed my concerns.

**Key Questions For Authors:**

See weaknesses.

**Limitations:**

Yes. The conclusion explicitly states that the guarantees rely on coverage and Bellman completeness, and identifies relaxing these assumptions and extending the analysis to broader settings as future work. However, the limitations discussion is still somewhat light. In particular, the paper could more directly emphasize the gap between the theory and practical deep DQN.

**Strengths And Weaknesses:**

**Strength:**
1. Unified theoretical perspective:
The use of the $\alpha$-condition to bridge the geometry mismatch is a clean way to unify different loss functions under one analytical umbrella

2. Reasonable motivation for soft-$\ell_\infty$:
The shift toward soft-$\ell_\infty$ is well-justified. Instead of just "trying a new loss," the authors show why it naturally reduces the dependence on the minimum sampling mass $\underline{w}$ that plagues MSE-based analysis.

**Weakness:**
1. The main results depend on small inherent Bellman error and a coverage assumption requiring full support with strictly positive lower bound $w_{\min}$. These are standard assumptions in fitted value-iteration style analyses, but in practical deep RL they function more as structural simplifications than as conditions that can be meaningfully checked.

2. The fitted-Q-iteration interpretation with frozen targets is a sensible abstraction, but it leaves aside many practical ingredients, including sampling noise, minibatch effects, the interaction between target refresh frequency and exploration, and the nonconvexity induced by deep parameterization. The paper treats the inner loop as a tabular proxy, which simplifies things significantly. I would like to see a more candid discussion on how these $\alpha$-conditions might realistically hold (or fail) in deep, non-convex parameter spaces.

3. In Proposition 3.5, the complexity analysis is tied to normalized sign/steepest descent, yet the experiments use AdamW. While the authors argue these are related, the theoretical guarantee doesn't strictly cover the actual optimizer used in the benchmarks.

---

> ### Author Rebuttal · Authors · 2026-03-31
>
> We thank the reviewer for the thoughtful and constructive feedback. We address each point below.
>
> **W.1: On the assumptions.**
>
> We first clarify that small inherent Bellman error $\varepsilon_\infty$ is not required for convergence: the contraction rate $((1+\gamma)/2)^t$ holds regardless of $\varepsilon_\infty$, which only parameterizes the asymptotic radius $R_{\text{sur}}$. Our contribution is to make this relationship surrogate-specific: under the same $\varepsilon_\infty$, soft-$\ell_\infty$ achieves the most favorable scaling. We will revise Theorem 3.3 to remove the misleading qualifier "small." A natural empirical proxy for varying $\varepsilon_\infty$ is model capacity: recent scaling studies for value-based RL (Fu et al., 2025) consistently show that larger networks improve value estimation, which our theory formally explains through the $\varepsilon_\infty \to R_{\text{sur}}$ mechanism.
>
> Full coverage ($w > 0$) is standard in the fitted Q-iteration literature (Szepesvári, C. (2022)) and can be relaxed to concentrability coefficients (Munos & Szepesvári, 2008) at the cost of replacing $\ell_\infty$ with a distribution-dependent norm. We chose the current formulation for clarity. Importantly, $w$'s role differs critically across surrogates: MSE and soft-Huber require $\alpha < O(w)$, so poor coverage can prevent contraction entirely, whereas soft-$\ell_\infty$ requires only $\alpha_{\text{LSC}} < (1-\gamma)/(1+\gamma)$, independent of $w$. This structural advantage is itself a contribution of the analysis.
>
> **W.2: On the tabular proxy and deep parameterization.**
>
> We appreciate the call for a more candid discussion. We will expand the limitations section to state: (i) population-level analysis is a necessary foundation but does not account for minibatch effects; (ii) tabular complexity captures intrinsic surrogate curvature, not full deep optimization difficulty; (iii) target refresh–exploration interaction is not modeled.
>
> Our $\alpha$-condition operates in prediction space, it asks whether $Q$-values moved closer to the target, regardless of the parameter-space trajectory. This separation is a feature of the surrogate framework: it isolates loss design from parameterization, enabling global convergence statements not restricted by local landscape properties.
>
> The parameterization does interact with surrogate choice through the composed Hessian $\nabla^2_\theta \ell= J^\top \nabla^2 L J + \sum_i [\nabla L]_{i}\nabla^2Q_i$.
>
> where $J(\theta) = \nabla_\theta Q_\theta$.The first term transmits surrogate curvature $\nabla^2 L$ (surrogate-specific, characterized
> in Lemma C.1) through $J(\theta)$ (shared across surrogates for the same network). The second term couples surrogate gradients $\nabla L$ (surrogate-specific) with network curvature $\nabla^2 Q_i$ (shared).The first term transmits surrogate curvature through the shared Jacobian $J(\theta)$, preserving the relative ranking from Lemma C.1. The second term is modulated by surrogate-specific gradients: unbounded for MSE, bounded for soft-Huber, concentrated for soft-$\ell_\infty$, creating a tradeoff our adaptive $\beta$ schedule navigates: small $\beta$ early for conditioning, increasing later while maintaining $\beta R \approx O(1)$ (Figure 12 validates this).
>
> **W.3: On sign descent vs. AdamW.**
>
> Proposition 3.5 analyzes normalized sign descent deliberately: it is the steepest-descent step in $\ell_\infty$ geometry (Balles et al., 2020), which more closely matches Adam's coordinate-wise rescaled updates than standard gradient descent in $\ell_2$geometry. Analyzing standard GD would yield predictions in $\ell_2$ curvature that are *less* predictive of AdamW behavior. Sign descent in $\ell_\infty$ geometry is thus the tightest available analytical proxy for AdamW.
>
> Empirically, the theory's primary prediction, that soft-$\ell_\infty$ most effectively reduces worst-case Bellman residual, is confirmed across all eight Atari games under AdamW (Figure 2). Proposition 3.5 characterizes intrinsic surrogate curvature rather than exact AdamW step counts; We will clarifying this scope in the revision.
>
> **Additional improvements.** In the revision, we extend the experimental evaluation with Dueling DQN experiments confirming surrogate effectiveness across architectures
>
> https://anonymous.4open.science/r/rebuttal-275E/dueling_returns_grid.png
>
> https://anonymous.4open.science/r/rebuttal-275E/dueling_max_abs_td_grid.png
>
>
> **References**
>
> Fu, Preston, et al. "Compute-Optimal Scaling for Value-Based Deep RL." NeurIPS, 2025
>
> Szepesvári, C. . Algorithms for reinforcement learning. Springer nature, 2022
>
> Munos, Rémi, and Csaba Szepesvári. "Finite-Time Bounds for Fitted Value Iteration." Journal of Machine Learning Research, 2008.
>
> Balles, L., Pedregosa, F., & Roux, N. L. The geometry of sign gradient descent. arXiv preprint 2020

---

> > ### Author Rebuttal · Reviewer_y4RJ · 2026-04-03
> >
> > Thanks for the responses. I will maintain my positive score.

---

> > > ### Author Response · Authors · 2026-04-03
> > >
> > > We sincerely thank the reviewer for confirming that all concerns
> > > have been fully resolved. The feedback on sign descent framing,
> > > deep parameterization, and assumption clarity substantially
> > > improved the paper.
> > >
> > > We note that the original review identified W.1–W.3 as the primary
> > > weaknesses motivating the current scores. As these have now been
> > > fully addressed, and the rebuttal additionally provides new
> > > Dueling DQN experiments across 8 games that were not in the
> > > original submission. We hope the reviewer finds
> > > that the current version of the paper merits a reassessment.

---

### Decision · Program_Chairs · 2026-04-30

**Decision:**

Reject

**Comment:**

The paper re-examine the loss used to train DQN.  The paper is motivated by that the Bellman optimality equation is a contraction in L_\infty norm, while practical algorithms uses L_2 (MSE) or L_1 (Huber) losses on the Bellman error.  They propose the soft L_\infty loss to better approximate the contraction property of the Bellman optimality equation, and provide empirical study for three kinds of losses.

While all reviewers are in favor of the paper, I have the concern that the theory developed in this paper lags behind the state of the art, and does not adequately explain the theoretical advantage of the proposed L_\infty loss.   In particular, the following paper

Jinglin Chen and Nan Jiang.  Information-Theoretic Considerations in Batch Reinforcement Learning.  2019

gives a convergence analysis for fitted Q-iteration with L_2 loss, under *strictly weaker* assumptions than Theorem 3.3.

Chen and Jiang's assumptions:
1. bounded inherent Bellman error in a norm weighted by offline distribution  (their Assumption  2 and 3 with F=G --- see their discussion on the F=G case)
2. bounded concentrability coefficient (their Assumption 1)

The submitted paper's assumptions:
1. bounded inherent Bellman error in L_\infty norm (Definition 3.1)   --- stronger than 1 above
2. sufficient coverage 1/w (Assumption 3.2)   --- stronger than 2 above
3. \alpha condition  (Eq.(8))   --- absent in Chen and Jiang

Specifically, requiring \alpha <= O(w) for L_2 loss in Theorem 3.3 is misleading: existing analysis by Chen and Jiang does not require this restrictive assumption.  Note that w <= 1/(S*A) by its definition (S: number of states, A: number of actions).  In usual RL with function approximation we have S = \infty and thus w=0.

So I fail to see how the theory advances our knowledge of DQN/Fitted Q-iteration and backup the choice of soft L_\infty loss.  The paper needs to have much deeper discussion on this related work and its follow-up works.  Unfortunately, I cannot recommend for acceptance at this stage.